inorganic chemistry

green rust, selenium, immobilization, kinetics, mechanism

**Author for correspondence:**
Chiharu Tokoro
e-mail: tokoro@waseda.jp

This article has been edited by the Royal Society of Chemistry, including the commissioning, peer review process and editorial aspects up to the point of acceptance.

# Kinetics and mechanism of selenate and selenite removal in solution by green rust-sulfate

Aina Onoguchi[1], Giuseppe Granata[1], Daisuke Haraguchi[2], Hiroshi Hayashi[2] and Chiharu Tokoro[1]

[1]Department of Resources and Environmental Engineering, Waseda University, Okubo 3-4-1, Shinjuku, Tokyo 169-8555, Japan
[2]Central Research Institute, Mitsubishi Materials Corporation, 15-2, Fukimatsu, Onahama, Iwaki, Fukushima 971-8101, Japan

CT, 0000-0001-6214-0402

This work investigated the removal of selenite and selenate from water by green rust (GR) sulfate. Selenite was immobilized by simple adsorption onto GR at pH 8, and by adsorption–reduction at pH 9. Selenate was immobilized by adsorption–reduction to selenite and zero valent selenium ($Se^0$) at both pH 8 and 9. In the process, GR oxidized to a mixture of goethite (FeOOH) and magnetite ($Fe_3O_4$). The kinetics of selenite and selenate sorption at the GR–water interface was described through a pseudo-second-order model. X-ray absorption spectroscopy data enabled to elucidate the concentration profiles of Se and Fe species in the solid phase and allowed to distinguish two removal mechanisms, namely adsorption and reduction. Selenite and selenate were reduced by GR through homogeneous solid-phase reaction upon adsorption and by heterogeneous reaction at the solid–liquid interface. The selenite reduced through heterogeneous reduction with GR was adsorbed onto GR but not reduced further. The redox reaction between GR and selenite/selenate was kinetically described through an irreversible second-order bimolecular reaction model based on XAFS concentration profiles. Although the redox reaction became faster at pH 9, simple adsorption was always the fastest removal mechanism.

## 1. Introduction

Selenium is an essential element for humans, but it is associated with acute toxicity above $1.4 \, \mathrm{mg \, l^{-1}}$ and with mutagenicity above 12 mM [1,2]. It commonly exists under four oxidation

states, among which Se(IV) as selenite ($SeO_3^{2-}$) and Se(VI) as selenate ($SeO_4^{2-}$) are the most stable species in water. Besides, Se(IV) is more toxic than Se(VI) [3]. Pollution by selenate and selenite occurs in both soils and water [4], arising mostly from anthropic activities such as agricultural practices [5] and coal combustion [4]. Mining activities also contribute to the contamination issue, especially when acid mine drainage dissolves the selenium naturally contained in ores/minerals [6,7]. These anthropogenic activities are carried out worldwide and involve large amounts of water. Al Kuisi & Abdel-Fattah [8], Kumar & Riyazuddin [9] and Bajaj et al. [10] well described the geography and the extent of environment contamination by selenium. Clearly, this issue should not be underestimated.

The removal of $SeO_4^{2-}$ and $SeO_3^{2-}$ from water has been the subject of many research works. Biological methods under anaerobic [11], oxic [12] and anoxic [13] conditions were proven effective and sustainable but relatively slow. Adsorption on carbon nanospheres [14], ferrous hydroxide [15], ettringite [16] and barite [17] were described as possible alternatives to biological methods. However, much research is still needed to achieve the efficiency and/or sustainability of reactive materials used in the clean-up of polluted environments. Reduction with zero-valent iron was also proven as an efficient method to reduce selenate and selenite to $Se^0$ [13,18–20]. However, large amounts of zero-valent iron (at least 3 molFe/molSe) are required to reduce and immobilize selenate.

An ideal alternative to zero valent iron is the immobilization on green rust (GR), when the contaminated water contains dissolved iron [21]. GR is a mixed Fe(II)–Fe(III) hydroxide compound with layered double hydroxide structure [22]. GR can be described through the formula $[Fe^{II}_{(1-x)}Fe^{III}_x(OH)_2]^{x+}[(x/n)A_n(mx/n)H_2O]_x$, where $A$ represents anions such as chloride, carbonate [23] or sulfate [24,25]. In GR, positively charged brucite-like layers of edge-sharing octahedrally coordinated Fe(II) and Fe(III) hydroxide units are intercalated with anionic species and water molecules [21,26]. Due to the presence of Fe(II) on the surface, GR exhibits a reducing potential that can be conveniently used to reduce oxidized contaminants such as nitrate [27], chromate [28] and arsenic anions [29,30]. The low-cost and easy production using novel methods [22] make GR a suitable removing agent for selenium species too.

GR also showed an interesting activity in the immobilization of selenate from water. Refait et al. [31] described the intercalation of selenate into GR's interlayer [21]. However, authors did not assess the reducing capability of GR and its fate upon intercalation. Hayashi et al. [30] highlighted that GR can adsorb and reduce selenate to $Se^0$, oxidizing to goethite (FeOOH) and/or magnetite ($Fe_3O_4$) depending on oxygen fugacity. Authors speculated about the simultaneous reduction of selenate to selenite and $Se^0$ but did not provide any specific evidence for this process [32]. Moreover, neither the reactions pathway nor the removal kinetics have been elucidated, while there is no information about the possibility to remove selenite.

This study addresses the removal of selenite and selenate at pH 8 and 9 by GR-sulfate. The slightly alkaline pH range was chosen based on the assumption that a neutralization operation would be preliminarily carried out to precipitate the other heavy metals contained in the wastewater. GR-sulfate was selected over other types of GRs, because the sulfate ion is often available in wastewaters contaminated by selenium and acid mine drainage. Batch experiments were carried out to investigate the fate of selenium and iron species and the removal kinetics. An approach based on inductively coupled plasma (ICP) and X-ray adsorption fine structure (XAFS) analysis was used to elucidate the fate of Se(IV), Se(VI) and Fe species, clarify the reactions pathway and assess the contribution of adsorption and reduction to the overall Se removal kinetics.

# 2. Material and methods

## 2.1. Synthesis of green rust

GR preparation was conducted as described by previous researchers [32]. All operations were performed inside a glove box (UN-650 L, Unico Ltd, Japan) under Ar atmosphere to prevent GR oxidation. All solutions were prepared using ultrapure water (Aquarius RFD 240NA, Advantec, Japan). All solutions were purged with Ar gas to remove dissolved oxygen prior to use. The GR precursor solution was prepared by dissolving $FeSO_4 \cdot 7H_2O$ and $Fe_2(SO_4)_3 \cdot nH_2O$ in the ratio of Fe(II)/Fe(III) = 0.75 with a total Fe concentration of $0.4 \text{ mol l}^{-1}$. The solution was then titrated with 8 M NaOH (addition speed: $2.3 \times 10^{-7} \text{ dm}^3 \text{ min}^{-1}$) up to pH 7.5 [19,20,28]. GR was separated from the obtained slurry by centrifugation (Himac CR21, Hitachi, Japan) at 1000 g for 10 min and washed with ultrapure water three times. After washing, the GR precipitate was re-suspended in de-oxygenated water, and later used as it was in removal experiments.

## 2.2. Selenium removal experiments

Removal experiments were conducted by adding the GR suspension to the aqueous solutions containing Se. Removal experiments were conducted by adding the GR suspension to the aqueous solutions containing Se. The suspension was mixed by magnetic stirring (300 r.p.m.) and purged continuously with Ar gas to limit the concentration of dissolved oxygen. The total volume was set to reach a total Fe concentration of $0.036\,mol\,l^{-1}$ $(2000\,mg\,l^{-1})$.

The Se concentration was set at $500\,mg\,l^{-1}$ while the pH was adjusted to 8 or 9 by adding 0.25 M NaOH. An automatic titrator (TS-2000, HIRANUMA SANGYO, Japan) adding 0.5 M NaOH was used to compensate for the pH decrease during the experiments. The redox potential was continuously monitored through an ORP electrode (D-75, HORIBA, Japan).

## 2.3. Analysis of solids and solutions

The concentration of Se in solution was determined by inductively coupled plasma atomic emission spectrometry (ICP-AES, SPS7800, Seiko Instrument, Japan) after filtration (0.1 µm, Whatman). The mineralogy and crystal structure of solid products from removal experiments were determined by X-ray diffraction (Geiger flex RAD-IX, Rigaku, Japan) with a copper target (CuKα) a crystal graphite monochromator and a scintillation detector. The X-ray source was operated at 40 kV and 30 mA with step scanning from $2\theta$ values of 2–80°, sequential increments of 0.02° and a scan speed of $2°\,min^{-1}$. To avoid the oxidation of samples between experiments and analysis, the collected paste samples were mixed with glycerol and transferred to the XRD chamber [33]. The concentration of Fe(II) was determined through the phenanthroline method and UV–Vis analysis (DR5400, Hach, USA). The total Fe(T-Fe) was determined by Phenanthroline method upon Fe(III) reduction with hydroxylammonium chloride. Size and morphology of solid products were examined using a transmission electron microscope (TEM-EDS, JEM2100, JEOL, Japan) equipped with an energy dispersive probe and operated at an accelerating voltage of 200 kV. Image mapping was applied to the same micrographs to identify the distribution of Fe and Se in the solid phase. For this purpose, samples were set on an elastic carbon support film (STEM 100 Cu grid, grid pitch 100 µm, Okenshoji, Japan) drying directly from the Ar-purged glove box after drying in vacuum. The Se K-edge and Fe K-edge X-ray adsorption fine structure (XAFS) were performed at the BL5S1 beamline of the Aichi Synchrotron Radiation Center (Aichi Science and Technology Foundation, Japan) using the transmission method from pelletized samples. The pellets for XAFS analysis were prepared through vacuum drying, grinding and mixing with boron nitride [34] in the Ar-purged glove box after solid–liquid separation of the slurry by filtration on 0.1 µm membrane filters. The Fe K-edge EXAFS function was derived from the raw Fe K-edge XAFS spectra through pre-edge and post-edge linear background subtraction and then normalization with respect to the edge jump. After being $k^3$-weighted, where $k$ is the photoelectron wavenumber, the Fe K-edge $k^3$ weighted EXAFS function was obtained. All XAFS analysis was performed using Athena.

# 3. Results and discussion

## 3.1. Removal experiments

The concentration of dissolved selenium in the removal experiments with Se(IV) and Se(VI) at pH 8 and 9 is shown in figure 1.

Results highlighted that selenite was removed to a larger extent at pH 8, whereas pH 9 produced the largest removal of selenate. At these pH values, selenate ($pK_a = 1.9$) was present in solution only as $SeO_4^{2-}$, while selenite ($pK_2$ 8.32) was present as both $SeO_3^{2-}$ and $HSeO_3^{-}$. Since all species were in their anionic form, the different removal between selenite and selenate could be explained by considering that different phenomena might contribute to the immobilization of Se species. In this view, a larger removal at low pH would be expected if the main removal mechanism was adsorption (lower competition with hydroxyl ions). By contrast, a larger removal at higher pH would be expected if the main removal mechanism was based on the redox reaction with GR (hydroxyl ions required). In the experiment with selenite at pH 8, the Se concentration dropped from the initial 500 to approximately $300\,mg\,l^{-1}$ upon contact with GR, and then reached a plateau in 30 min. On the other hand, a re-dissolution of Se followed the initial concentration drop from 500 to $350\,mg\,l^{-1}$ at pH 9. After this re-dissolution, the concentration of Se in solution started decreasing again at a clearly lower rate. This trend suggested that

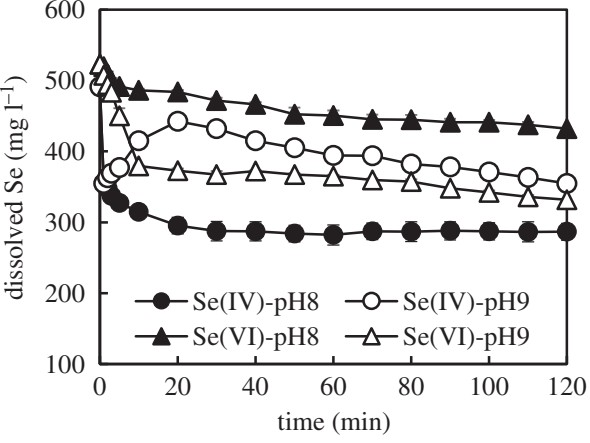

**Figure 1.** Se concentration in the removal experiments at pH 8 and 9.

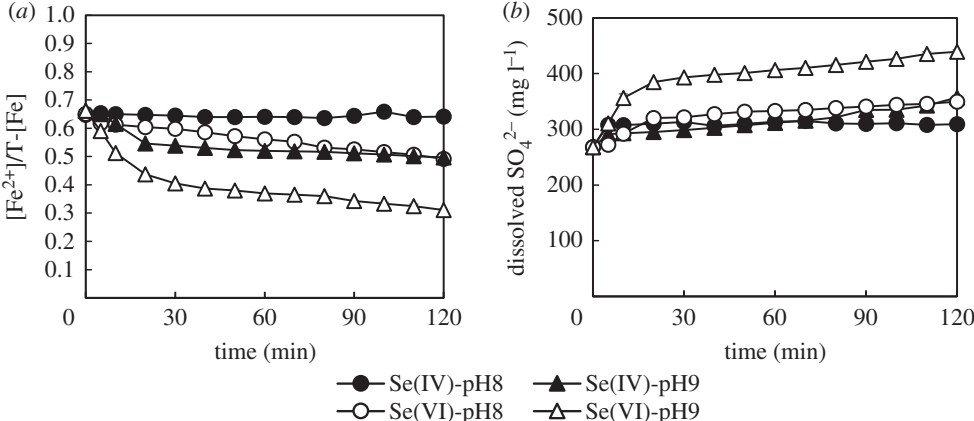

**Figure 2.** [Fe(II)]/[T-Fe] molar ratio (*a*) and $SO_4^{2-}$ release (*b*) during removal experiments.

at the beginning of the experiment selenate was quickly adsorbed onto GR. Following this adsorption, a part of the adsorbed Se(VI) was reduced to Se(IV) and released into solution. In the experiments with Se(VI), Se concentration exhibited a similar profile but with a smoother initial decrease. Furthermore, more Se could be removed at pH 9. The larger removal of selenate at pH 9 contrasts with what was reported by Hayashi *et al.* [30], who described pH 7.5 as the best condition to remove Se(VI) in the pH range 7–10. To elucidate this aspect, the trends of sulfate concentration and $Fe^{2+}$/total Fe molar ratio ($Fe^{2+}$/T-Fe) in solution were continuously monitored during the experiments. Results are shown in figure 2*a,b*. Figure 3 shows the redox potential (versus SHE) measured during removal experiments and the cumulative addition of NaOH to titrate the system.

The initial $Fe^{2+}$/T-Fe ratio in GR was 0.66, very close to the ideal ratio reported in the literature [35,36]. This ratio progressively decreased in all experiments, besides the one with Se(IV) at pH 8. The decreasing $Fe^{2+}$/T-Fe ratio was a clear suggestion that GR was being oxidized while removing selenite and/or selenate. Accordingly, $SO_4^{2-}$ was released in solution following GR oxidation (figure 2*b*). Both trends were in general more pronounced at pH 9. By contrast, the $Fe^{2+}$/T-Fe ratio was nearly constant in the experiment with selenite at pH 8, though the removal of selenium was the largest. This evidence revealed that selenite at pH 8 was removed from solution without reduction. The very low amount of sulfate released in the same experiment can be considered as evidence that selenite was removed by simple adsorption onto GR. In fact, if the immobilization occurred via intercalation in GR interlayer, the sulfate release should be stoichiometric to selenite in order to maintain the charge balance within GR. If the different pH-dependent behaviour can be explained considering that higher pH favour more the redox reactions, the different behaviour of selenite and selenate at the same pH (pH 8) must be explained considering how prone these two species are to adsorption onto GR. Indeed, selenate is smaller than selenite and more negatively charged because completely deprotonated at pH 8. Therefore, a larger removal of selenite by simple adsorption could be somehow expected. By contrast, the reduction

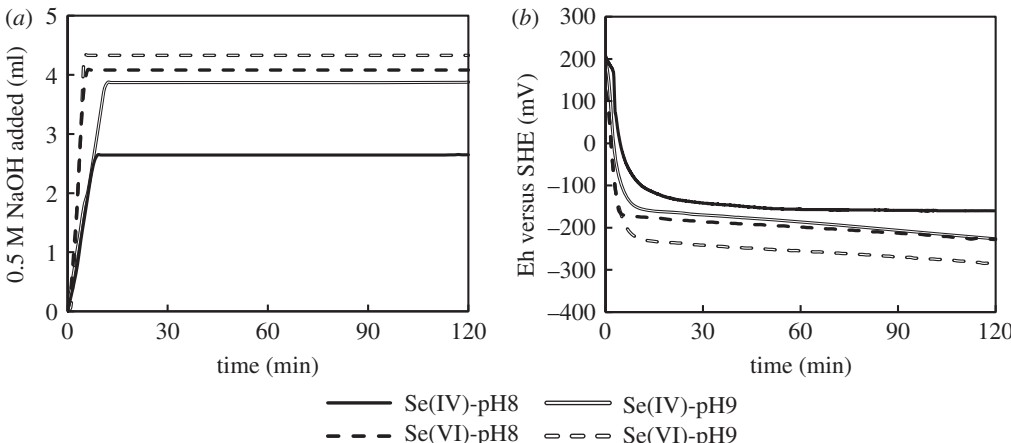

**Figure 3.** Redox potential versus volume of 0.5 M NaOH added (*a*) and SHE (*b*) during the removal experiments.

of selenate at the same pH was more pronounced, as highlighted by the larger decrease of $Fe^{2+}$/T-Fe ratio and sulfate release. This evidence could be reasonably explained considering the thermodynamic advantage associated with the reduction of more compound. Like in the experiments at pH 8, the concentration of dissolved Se decreased rapidly upon contact with GR also at pH 9, thus suggesting a quick adsorption onto GR. However, at pH 9 selenite re-dissolved again after the initial removal. Given the decreasing $Fe^{2+}$/T-Fe ratio and the release of $SO_4^{2-}$, it is possible that about 50% of the selenite initially adsorbed was released due to GR oxidation.

The above-mentioned consideration was supported also by the redox potential in solution (figure 3*b*). In all experiments, the redox potential sharply decreased upon adding GR to the selenite/selenate solutions. The polarization at lower potentials revealed as the system became electrochemically controlled by the redox couples $Fe^{3+}$/$Fe^{2+}$ and $SeO_4^{2-}$/$SeO_3^{2-}$. Following the initial sharp polarization, the potential started decreasing again but slowly due to the reduction of selenite/selenate. The slow decrease was not observed in the experiment with Se(IV) at pH 8, as a further proof that selenite was removed via simple adsorption. By contrast, the largest Eh decrease was observed in the experiments with selenate at pH 9, as also expected from the $Fe^{2+}$/T-Fe trend. This evidence was a further indication that (i) selenate was easier to reduce than selenite, and that (ii) pH 9 favoured the redox reaction.

## 3.2. Transformation of GR and Se species

The XRD patterns of the solid residues from removal experiments are shown in figure 4.

The XRD patterns in figure 4 exhibited the typical peaks of goethite and magnetite under all investigated conditions except for Se(IV) at pH 8. Although selenium species were not detected, the presence of only GR in the XRD pattern of the solid residue from the experiment with selenite at pH 8 confirmed the removal via adsorption. The presence of Se in all residual solids was confirmed by FE-TEM-EDX analysis and image mapping (figure 5). In the TEM micrographs, the hexagonal flakes corresponded to GR [26] while the rods observed in the other experiments could be either magnetite or goethite, in accordance with XRD results. The solid residue from the experiment with selenite at pH 9 exhibited yellow spherical particles, most likely corresponding to $Se^0$. On the other hand, the solid residues from the experiments with selenate exhibited also yellow portions with elongated shape. This result suggests that some of the selenite produced from the reaction between selenate and GR was adsorbed onto the as-formed magnetite and/or goethite.

## 3.3. XAFS results

XAFS analysis was conducted in the Se K-edge and Fe K-edge to identify Fe and Se species during removal experiments [37]. The Se K-edge normalized XANES spectra and the Fe K-edge $k^3$ weighted EXAFS spectra of solid residues from removal experiments are shown in figure 6*a,b*. Results of peak fitting against reference materials are listed in table 1.

The Se K-edge XANES and the EXAFS function $k^3\chi(k)$ spectra of the solid residue from the experiment with Se(IV) at pH 8 revealed only the presence of selenite and GR, as also expected from

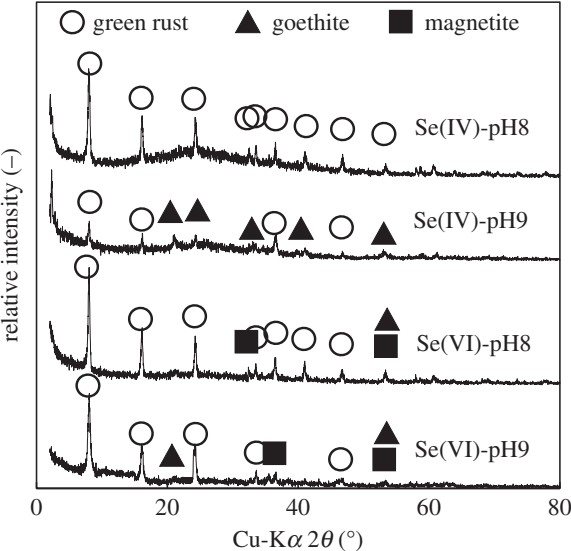

**Figure 4.** XRD patterns of the solid residues from removal experiments.

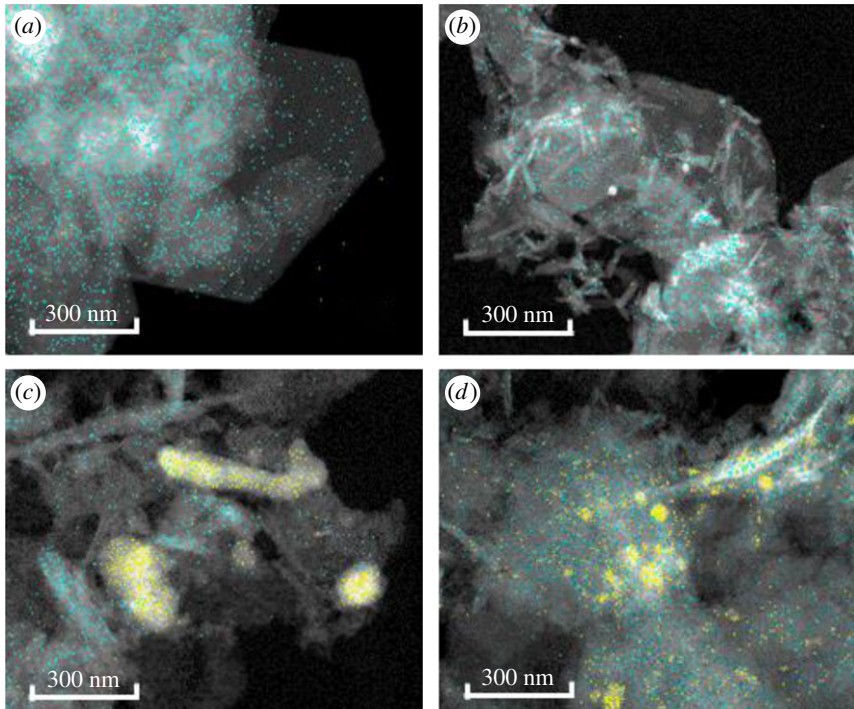

**Figure 5.** TEM micrographs and image mapping of residual solids from experiments with Se(IV) at pH8 (*a*), Se(IV) at pH9 (*b*), Se(VI) at pH8 (*c*), Se(VI) at pH9 (*d*). (Blue, Fe; Yellow, Se).

the above-described results. By contrast, Se(IV), $Se^0$, GR and goethite were confirmed in the solid residue at pH 9. Fitting results highlighted that about 25% of selenite was reduced to $Se^0$ at the expenses of GR, which was oxidized to goethite as shown below

$$Fe_4Fe_2(OH)_{12}SO_4 + SeO_3^{2-} \rightarrow 6FeOOH(s) + SO_4^{2-} + Se^0 + 3H_2O. \tag{3.1}$$

XAFS results from the experiment with selenate at pH 8 exhibited multiple peaks around 12 652, 12 657 and 12 659 eV. These peaks revealed the presence of $Se^0$ and Se(IV) in the solid residue. The EXAFS function $k^3\chi(k)$ of the same sample revealed the conversion of about 29% GR to 22% goethite and

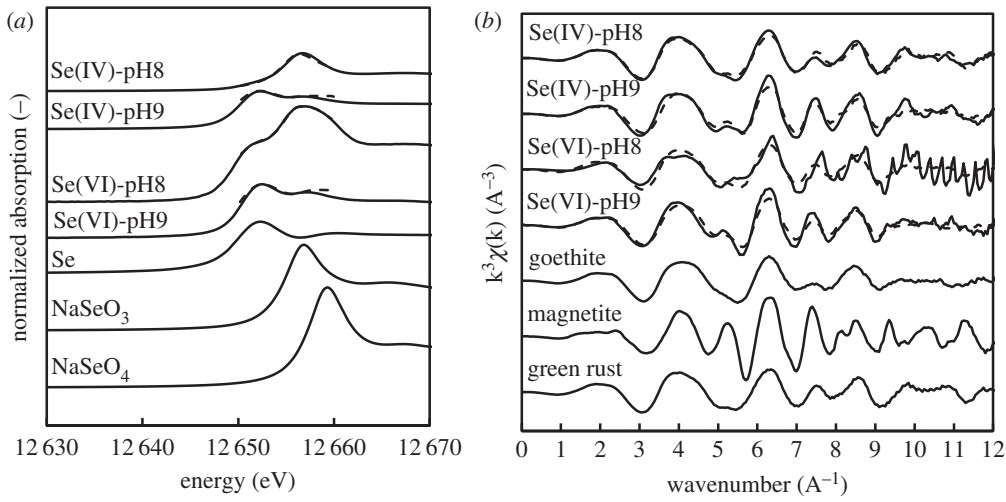

**Figure 6.** Normalized Se K-edge XANES spectra (*a*) and Fe K-edge $k^3$ weighted EXAFS spectra (*b*) of end-products against $Se^0$, $Na_2SeO_3$, $Na_3SeO_4$, GR, goethite and magnetite reference materials.

**Table 1.** Concentration of Se and Fe species in end-products based on XAFS fitting.

| condition | Se(VI) (%) | Se(IV) (%) | $Se^0$ (%) | GR (%) | goethite (%) | magnetite (%) |
|---|---|---|---|---|---|---|
| Se(IV) pH 8 | — | 100 | — | 100 | 0 | 0 |
| Se(IV) pH 9 | — | 8.9 | 91.1 | 73.1 | 26.9 | 0 |
| Se(VI) pH 8 | 0 | 55.4 | 44.6 | 70.5 | 22.3 | 7.2 |
| Se(VI) pH 9 | 0 | 47.1 | 52.9 | 17.7 | 64.0 | 18.3 |

about 7% magnetite. Clearly, selenate was reduced to selenite and $Se^0$ using the electrons generated from the oxidation of GR to goethite and magnetite as in (3.2)–(3.5).

$$3Fe_4Fe_2(OH)_{12}SO_4 + 2SeO_4^{2-} + 2OH^- \rightarrow 18FeOOH(s) + 3SO_4^{2-} + 2Se^0 + 10H_2O, \quad (3.2)$$

$$Fe_4Fe_2(OH)_{12}SO_4 + 2SeO_4^{2-} + 2OH^- \rightarrow 6FeOOH(s) + SO_4^{2-} + 2SeO_3^2 + 4H_2O, \quad (3.3)$$

$$3Fe_4Fe_2(OH)_{12}SO_4 + SeO_4^{2-} + 4OH^- \rightarrow 6Fe_3O_4(s) + 3SO_4^{2-} + Se^0 + 20H_2O \quad (3.4)$$

and

$$Fe_4Fe_2(OH)_{12}SO_4 + SeO_4^{2-} + 2OH^- \rightarrow 2Fe_3O_4(s) + SO_4^{2-} + SeO_3^{2-} + 7H_2O. \quad (3.5)$$

The presence of Se(IV) in the solid residues revealed as the selenite formed from the reduction of selenate was adsorbed by GR and/or iron products. Nevertheless, it was not possible to understand whether selenite was adsorbed upon heterogeneous reduction between GR and selenate, or if it was generated from already adsorbed selenate that remained adsorbed after reduction. This aspect will be discussed in §3.4.

Increasing the removal pH from 8 to 9 resulted in a more pronounced reduction of selenate to $Se^0$. As a consequence of the larger amount of exchanged electrons, more GR oxidized to goethite (64%) and magnetite (18%). The significantly positive effect of pH on the extent of redox reaction is consistent with the stoichiometry of reactions (3.2)–(3.5) as hydroxyl ions are required for the reaction to take place. The simultaneous reduction of selenate to both goethite and magnetite contrasts with results reported by Hayashi *et al.* [30] and Myneni *et al.* [37], who described the preferential formation of magnetite at pH 9 [38]. The difference might be due to the different experimental conditions used in this work (e.g. constant pH through addition of NaOH, lower Fe/Se molar ratio).

## 3.4. Reaction pathway and kinetics

Since the immobilization of selenium involved the adsorption and/or reduction of Se species on GR surface, the kinetic analysis of Se removal was performed using a pseudo-second-order model [39]. According to the model, the adsorption rate is proportional to the second power of the surface

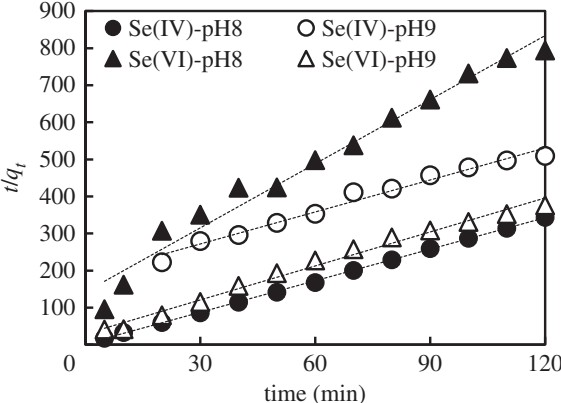

**Figure 7.** Fitting of results from removal experiments with pseudo-second-order kinetic model.

**Table 2.** Fitting parameters for the lines in figure 7 (the rate constant was converted from g-GR $\times$ mmol-Se$^{-1}$ $\times$ min$^{-1}$ to l $\times$ mmol-Se$^{-1}$ $\times$ min$^{-1}$).

| condition | $q_e$ (mmol g$^{-1}$) | kpll (l $\times$ mmol-Se$^{-1}$ $\times$ min$^{-1}$) | $R^2$ |
|---|---|---|---|
| Se(IV) pH 8 | $2.3 \times 10^{-1}$ | $9.9 \times 10^{-1}$ | 1.000 |
| Se(IV) pH 9 | $1.4 \times 10^{-1}$ | $1.2 \times 10^{-2}$ | 0.980 |
| Se(VI) pH 8 | $9.6 \times 10^{-2}$ | $4.7 \times 10^{-2}$ | 0.974 |
| Se(VI) pH 9 | $2.0 \times 10^{-1}$ | $8.6 \times 10^{-2}$ | 0.989 |

available for adsorption/reaction [40] as shown below

$$\frac{dq_t}{dt} = k(q_e - q_t)^2, \tag{3.6}$$

where $q_t$ and $q_e$ are the sorption densities at time $t$ and equilibrium, determined, respectively, as shown below

$$q_e = \frac{(C_i - C_e)V}{M} \tag{3.7}$$

and

$$q_t = \frac{(C_i - C_t)V}{M}, \tag{3.8}$$

where $C_i$, $C_e$ and $C_t$ (mmol l$^{-1}$) are the concentrations of Se(IV) or Se(VI) at initial, equilibrium and time $t$, respectively, $M$ is the mass of T-Fe (g) calculated from the formula of GR, $V$ is the volume of solution (l) and $k$ is the adsorption rate constant. For the boundary conditions $t = 0$ to $t = t$ and $q_t$, the integrated form of equation (3.6) becomes as shown below

$$\frac{t}{q_t} = \frac{1}{q_e}t + \frac{1}{q_e^2 k}. \tag{3.9}$$

Accordingly, if the removal reaction follows a pseudo-second-order kinetics, the experimental points plotted as $t/q_t$ versus time align through a straight line with slope $1/q_e$ and intercept $1/q_e^2 k$. The $k$ in formula (3.9) is shown as kpII in table 2. The plots of $t/qt$ versus time are shown in figure 7. Fitting parameters from figure 7 are listed in table 2.

The good fitting from figure 7 highlighted the suitability of the pseudo-second-order model to describe the removal of selenite and selenate by GR at pH 8 and 9. The removal rate constant was the highest in the experiment with selenite at pH 8, thus highlighting that simple adsorption was the fastest removal mechanism. Increasing the pH from 8 to 9 promoted the redox reaction between selenite and GR. As a consequence, the kinetics was slower because GR structure was destroyed and the adsorbed selenite was released in solution. By contrast, the removal of selenate was the fastest at pH 9, the condition that determined the most pronounced redox reaction.

**Table 3.** Fitting parameters for the lines in figure 10.

| condition | $k_{rdx}$ (l $\times$ mmol$^{-1}$ $\times$ min$^{-1}$) | $R^2$ |
|---|---|---|
| Se(IV) pH 8 | — | — |
| Se(IV) pH 9 | $5.0 \times 10^{-4}$ | 0.990 |
| Se(VI) pH 8 | $7.4 \times 10^{-4}$ | 0.999 |
| Se(VI) pH 9 | $6.3 \times 10^{-3}$ | 0.984 |

To identify the phenomena involved in the removal of selenium and to assess their contribution to the overall kinetics, XAFS analysis was extended to samples at 5, 15, 20, 30, 90 and 120 min. Fitting results from XAFS spectra are listed in electronic supplementary material, table S1 while the plots of data in table 3 representing the evolution of Se and Fe species through time are shown in figure 8. The XAFS spectra are provided as electronic supplementary material, figures S1 and S2.

From the graphs in figure 8, it was clear that selenite was adsorbed onto GR and/or reduced to Se$^0$ upon adsorption. The concentration profiles also suggested a faster adsorption followed by slower redox reactions, where Se(IV) concentration decreased at the advantage of Se$^0$ while Se(VI) was reduced to Se(IV) and Se$^0$. The first interesting point to note is the concentration profile of Se(VI) in the solid phase. Although for practical reasons the first sample for XAFS was taken 5 min after realizing the contact Se-GR, the Se(VI) concentration must be clearly 0 before contact (time 0). Therefore, the concentration of Se(VI) must have reached a maximum at $t \leq 5$ min before being consumed. This was clear evidence that selenate was quickly adsorbed onto GR, as initially suggested by figure 1. Following the adsorption, Se(VI) was reduced to Se(IV) and/or Se$^0$. However, figure 8$b$–$f$ revealed also that the concentrations of reduced selenium species (Se$^0$ in the experiment with selenite, Se(IV) and Se$^0$ in the experiment with selenate) were still decreasing after all initially adsorbed Se was consumed. This evidence suggested two considerations. In the first place, not only the adsorbed selenite/selenate could be reduced by GR (homogeneous reaction) but also their dissolved counterparts upon contact with GR (heterogeneous reaction). Moreover, because selenite is soluble, the presence of Se(IV) in the solid phase could be explained only considering the adsorption of the selenite formed by heterogeneous reaction between GR and selenate. In this view, the slow removal of Se that followed the initial large concentration drop observed in figure 1 must be the main consequence of the heterogeneous redox reaction. Another interesting point to note is the simultaneous reduction of selenate to selenite and Se$^0$ (figure 8$d$–$f$). Although the reduction of selenate could be described by a series of two irreversible reductions (e.g. to selenite first and, in turn, to Se$^0$), the monotonic trend of Se(IV) was more consistent with a parallel reduction of the adsorbed selenate to adsorbed selenite and/or Se$^0$. In other words, the adsorbed selenite could not be reduced further to Se$^0$. This evidence might be caused by the depletion of Fe(II) in the proximity of the Fe(III) adsorption sites due to the reduction of selenate to selenite. The concentration profiles of Fe species also suggested a similar consideration. The monotonic trends of GR, magnetite and goethite suggest that GR oxidized either to magnetite or to goethite, whereas the oxidation of magnetite to goethite did not occur.

Based on GR consumption (XAFS results), amount of precipitated iron, selenium removed (ICP results) and the stoichiometry of redox reactions (3.2)–(3.5), an electron balance was carried out to assess the efficiency of GR to reduce Se(IV) and Se(VI). The relationship between electrons produced through GR oxidation and electrons used in the reduction of Se(IV) and Se(VI) is shown in figure 9. The linear correlation with unitary slope shown in figure 9 confirmed the effectiveness of GR in the reduction of selenite at pH 9 and selenate at both pH 8 and 9. The slight positive deviation observed from 0.010 mol dm$^{-3}$ (condition corresponding to the experimental points at 90 and 120 min) in the experiments with Se(VI) at pH 9 revealed that some of the electrons from GR were not used to reduce selenate. There are two possible explanations for the observed deviation. One is the unwanted oxidation of GR by air, though Ar gas was continuously bubbled into the reacting system. Another possible explanation involves the reduction of selenate to soluble species such as selenite or selenide [41] that were not adsorbed.

Given the multiple phenomena occurring simultaneously in the investigated GR-Se systems (e.g. simple adsorption, homogeneous/heterogeneous redox reaction), it would be interesting to assess the relationship between them and their contribution on the overall Se removal and kinetics. For this

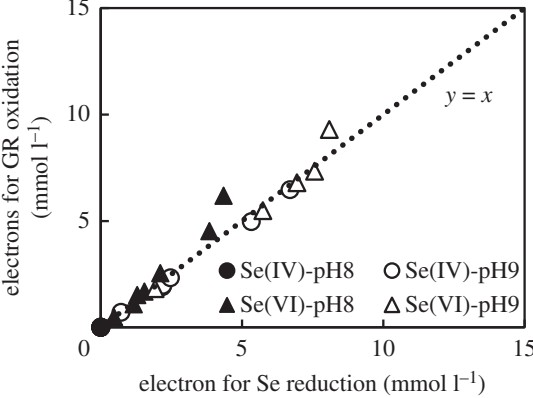

**Figure 8.** Concentration profiles of Fe and Se species in the solid phase for the experiments with Se(IV) at pH 9 (*a,b*), Se(VI) at pH 8 (*c,d*) and Se(VI) at pH 9 (*e,f*).

**Figure 9.** Correlation between electrons for GR oxidation and electrons for Se(IV)/Se(VI) reduction.

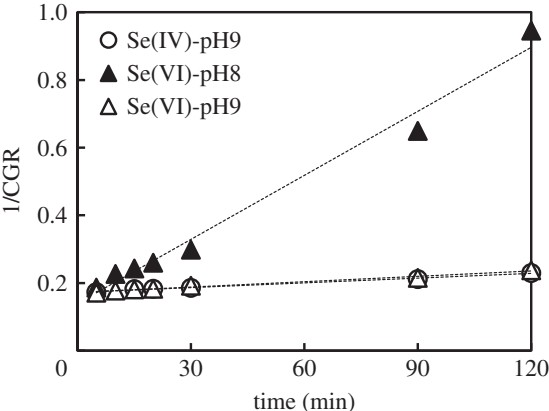

**Figure 10.** Fitting of GR concentration profiles from XAFS results with second-order kinetic model.

purpose, the kinetics of the redox reaction between GR and selenite/selenate were compared with the above-described pseudo-second-order describing the overall removal kinetics. In this work, the kinetics of the redox reactions was determined based on GR concentration profiles from XAFS analysis (figure 8). GR concentration profiles were preferred over the concentration profiles of Se(IV) and Se(VI) since the latest ones depend upon adsorption phenomena.

The reaction between GR and selenite/selenate was assumed to be a bimolecular first-order reaction to both GR and selenite/selenate, thus an overall second-order reaction of the type

$$A + B \rightarrow \text{Products.} \tag{3.10}$$

Unlike previous research works where the concentration of GR was set in large stoichiometric excess to the targeted species [42], in this work GR and selenite/selenate were used in the stoichiometric ratio. Under this condition, the second-order reaction in (3.10) can be conveniently treated as a second-order reaction to GR [43]. Accordingly,

$$\frac{\mathrm{d}C_{\mathrm{GR}}}{\mathrm{dt}} = k_{\mathrm{rdx}}C_{\mathrm{GR}}^2. \tag{3.11}$$

Integration of the second-order differential equations in (3.11) yields as follows:

$$\frac{1}{C_{\mathrm{GR}}} = \frac{1}{C_{\mathrm{GR0}}} + k_{\mathrm{rdx}}t, \tag{3.12}$$

where $C_{\mathrm{GR0}}$ and $C_{\mathrm{GR}}$ (mmol l$^{-1}$) are the concentrations of GR at time 0 and time $t$, while $k_{\mathrm{rdx}}$ (L $\times$ mmol$^{-1}$ $\times$ min$^{-1}$) is the rate constant of the second-order redox reaction between GR and selenite/selenate. Fitting results are shown in figure 10. Fitting parameters from figure 10 are listed in table 3.

The good correlation of the fitting lines in figure 10 confirmed the suitability of the overall second-order reaction model to describe the reduction of Se species by GR. Fitting results in table 3 confirmed as the redox reaction between GR and selenium species became faster at pH 9. The difference between the pseudo-second-order removal kinetics in table 3 and the second-order redox reaction rate constants in table 3 highlighted that simple adsorption was always the fastest removal mechanism. Comparing the rate parameters also suggested that the immobilization of selenium became slower after GR oxidized. Although the redox reactions are slower than adsorption and do not affect significantly the overall removal kinetics, knowing their rate parameters could be important for the management of end-products. Longer contact times would promote the formation of more crystalline solids (e.g. goethite, Se$^0$) exhibiting a better solid–liquid separability [44]. By contrast, shorter times would lead to a presumably weaker immobilization of selenate by simple adsorption. As a consequence, the selenium immobilized for shorter reaction times could be released more easily from the disposed sludge.

## 3.5. Discussion

The immobilization of selenium species by GR occurred through different phenomena depending on pH and Se species. In general, selenite could be removed from water to a larger extent than selenate. Simple adsorption was found to be the favoured removal mechanism at pH 8, whereas increasing the pH to 9

promoted the redox reaction between GR and Se species. This was evident already from the trends of $[Fe^{2+}]/[T-Fe]$, and it was later confirmed by XAFS analysis. All results indicated that selenite was immobilized by simple adsorption at pH 8 and by adsorption–reduction to $Se^0$ at pH 9. On the other hand, the removal of selenate involved the redox reaction with GR at both pH 8 and 9, though the redox reaction was more pronounced at pH 9. In the process, GR oxidized mainly to goethite but also to magnetite, while selenate was reduced to selenite and $Se^0$. The concentration profiles from XAFS results along with the electron balance suggested that the redox reaction between GR and Se species occurred after adsorption (homogeneous redox reaction) and without adsorption (heterogeneous redox reaction), with simple adsorption being always the fastest removal mechanism. The concentration profiles from XAFS analysis also suggested that the adsorbed selenate was reduced to selenite and/or $Se^0$ through a parallel reduction pathway. In other words, the adsorbed selenite could not be reduced further to $Se^0$, probably as a consequence of the depletion of Fe(II) in the proximity of the Fe(III) adsorption sites. Similarly, GR was reduced preferentially to goethite but also to magnetite, whereas the oxidation of magnetite to goethite did not occur.

## 4. Conclusion

This study confirmed the ability of GR to remove selenite and selenate from water, and provided useful information towards the elucidation of the removal mechanism. The immobilization of selenium species by GR occurred through different phenomena depending on pH and Se species. In general, pH 8 favoured the immobilization of selenium species via simple adsorption, whereas pH 9 favoured the redox reaction with GR. When the redox occurred, goethite was always the main oxidation product of GR. The kinetic analysis based on ICP and XAFS results highlighted that simple adsorption was always the fastest removal mechanism.

Although the redox reaction affects Se removal and removal kinetics to a lesser extent than adsorption, the contact time between GR and selenite/selenate shall be increased to obtain a more efficient immobilization of Se species and the formation of more crystalline products.

Data accessibility. Data available from the Dryad Digital Repository at: https://doi.org/10.5061/dryad.nk43478 [44].

Authors' contributions. A.O. gave essential contribution to the experimental acquisition of data and to the drafting of the article. G.G. gave substantial contribution to the design of the experiments, analysis and interpretation of data, drafting and revision of the article. C.T. gave substantial contribution to conception of the work, interpretation of data and revision of the article. All authors approve the final version and agree to be accountable for all aspects related to this work.

Funding. Authors also wish to thank Mitsubishi Material Corporation (MMC, Japan) for the support received during this research.

Competing interests. We have no competing interests.

Acknowledgements. A part of the present work was performed within the activities of Research Institute of Sustainable Future Society, Waseda Research Institute for Science and Engineering, Waseda University. The XAFS analysis was performed using the BL5S1 beamline at the Aichi Synchrotron Radiation Center, Aichi Science & Technology Foundation, Aichi, Japan (Proposal no. 2017P1001).

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
