## [Reviewer comments · Royal Society Open Science]

Review History

RSOS-181464.R0 (Original submission)

Review form: Reviewer 1

Is the manuscript scientifically sound in its present form?

No

Are the interpretations and conclusions justified by the results?

Yes

Is the language acceptable?

No

Is it clear how to access all supporting data?

Yes

Do you have any ethical concerns with this paper?

No

Have you any concerns about statistical analyses in this paper?

No

Recommendation?

Reject

Comments to the Author(s)

Text annotations are provided with the attached PDF file (Appendix A).

Review form: Reviewer 2

Is the manuscript scientifically sound in its present form?

Yes

Are the interpretations and conclusions justified by the results?

Yes

Is the language acceptable?

Yes

Is it clear how to access all supporting data?

Yes

Do you have any ethical concerns with this paper?

No

Have you any concerns about statistical analyses in this paper?

No

Recommendation?

Accept with minor revision (please list in comments)

Comments to the Author(s)

The manuscript "Kinetics and mechanism of selenate and selenite removal in solution by sulfate-green rust" (RSOS-181464) presents interest study about the Se removal using sulfate-green rust. I would say this study present high quality data to support their conclusion. The manuscript is generally well written and easy to understand. Since the adsorption and reduction of Se by GRs might be complicated, the author tried to make sufficient discussion to reveal the Se removal mechanism. I think this paper could be published with minor revision.

Specific Comments:

1. Intorduction P2, L1-2: "When the contaminated water contains already dissolved iron, an ideal alternative to externally-added removing agents is the immobilization on green rust". I think there should be other reasons to use GRs, such as the easy production using novel methods (reference 22) , cheap price but high reducing capacity.
2. Preparation of Green Rust: please add some references to support the synthesis methods.
3. Is it possible to merge Fig. 1 and Fig. 2 in one column? They can share the same X axis.
4. P4, L55: "selenate was easier to reduce than selenite", "that" should be "than"?
5. Fig 8 seems to repeatedly present the data in Table 3.

6. Fig. 10 should be removed since the author provides data in Table 4.

7. I think the authors have present sufficient data in this study, however, the discussion could be improve by separating the results and discussion, so that the highlights can be emphasized in the discussion. It might be a good way to make the logic clearer when the authors have much data to present.

By

Weizhao Yin
Associate Professor
Jinan University
Guangzhou 510006, China
weizhaoyin@jnu.edu.cn

Decision letter (RSOS-181464.R0)

06-Nov-2018

Dear Professor Tokoro:

Manuscript ID: RSOS-181464

Title: "Kinetics and mechanism of selenate and selenite removal in solution by sulfate-green rust"

Thank you for submitting the above manuscript to Royal Society Open Science. Your paper was sent to reviewers and their comments are included at the bottom of this letter.

In view of the concerns raised by the reviewers, the manuscript has been rejected in its current form. However, a new manuscript may be submitted which takes into consideration these comments.

Please note that resubmitting your manuscript does not guarantee eventual acceptance, and that your resubmission will be subject to peer review before a decision is made.

Your resubmitted manuscript should be submitted by 06-May-2019. If you are unable to submit by this date please contact the Editorial Office.

On behalf of the Subject Editor Professor Anthony Stace and the Associate Editor Dr Ya-Wen Wang

REVIEWER(S) REPORTS:

Associate Editor Comments to Author ():

RSC Associate Editor:

Comments to the Author:

(There are no comments.)

RSC Subject Editor:

Comments to the Author:

(There are no comments.)

Reviewers' Comments to Author:

Reviewer: 1

Comments to the Author(s)

Text annotations are provided with the attached PDF file.

Reviewer: 2

Comments to the Author(s)

The manuscript "Kinetics and mechanism of selenate and selenite removal in solution by sulfate-green rust" (RSOS-181464) presents interest study about the Se removal using sulfate-green rust. I would say this study present high quality data to support their conclusion. The manuscript is generally well written and easy to understand. Since the adsorption and reduction of Se by GRs might be complicated, the author tried to make sufficient discussion to reveal the Se removal mechanism. I think this paper could be published with minor revision.

Specific Comments:

1. Introduction P2, L1-2: "When the contaminated water contains already dissolved iron, an ideal alternative to externally-added removing agents is the immobilization on green rust". I think there should be other reasons to use GRs, such as the easy production using novel methods (reference 22), cheap price but high reducing capacity.
2. Preparation of Green Rust: please add some references to support the synthesis methods.
3. Is it possible to merge Fig. 1 and Fig. 2 in one column? They can share the same X axis.
4. P4, L55: "selenate was easier to reduce than selenite", "that" should be "than"?
5. Fig 8 seems to repeatedly present the data in Table 3.
6. Fig. 10 should be removed since the author provides data in Table 4.
7. I think the authors have present sufficient data in this study, however, the discussion could be improve by separating the results and discussion, so that the highlights can be emphasized in the

discussion. It might be a good way to make the logic clearer when the authors have much data to present.

By
Weizhao Yin

Associate Professor
Jinan University
Guangzhou 510006, China
weizhaoyin@jnu.edu.cn

Author's Response to Decision Letter for (RSOS-181464.R0)

See Appendices B & C.

RSOS-182147.R0

Review form: Reviewer 1

Is the manuscript scientifically sound in its present form?

Yes

Are the interpretations and conclusions justified by the results?

Yes

Is the language acceptable?

Yes

Is it clear how to access all supporting data?

Yes

Do you have any ethical concerns with this paper?

No

Have you any concerns about statistical analyses in this paper?

No

Recommendation?

Accept with minor revision (please list in comments)

Comments to the Author(s)

This resubmitted article by Onoguchi et al, describes the removal of selenite and selenate through processes involving adsorption or reduction by green rust.

The authors have done a good job improving the language and addressing major reviewer comments. I only have one comment:

Line 55-56: "The total volume was set to reach a total Fe concentration of 0.036 mol/L (2000 mg/L). The Se concentration was set at 500 mg/L while the pH was adjusted to 8 or 9 by adding 0.25 M NaOH. While the synthesis of GR was conducted inside a glovebox, removal experiments were conducted in a separable flask under open air atmosphere, for 2 hours under magnetic stirring (300 rpm) and constant pH. During the experiments, the suspensions were continuously purged with Ar gas to limit the concentration of dissolved oxygen"

I suggest the following alternative text to resolve the confusion with your 'open air' statement:

Removal experiments were conducted by adding the GR suspension to the aqueous solutions containing Se. The suspension was mixed by magnetic stirring (300 rpm) and purged continuously with Ar gas to limit the concentration of dissolved oxygen. The total volume was set to reach a total Fe concentration of 0.036 mol/L (2000 mg/L).

The Se concentration was set at 500 mg/L while the pH was adjusted to 8 or 9 by adding 0.25 M NaOH. An automatic titrator (TS-2000, HIRANUMA SANGYO, Japan) adding 0.5 M NaOH was used to compensate for the pH decrease during the experiments. The redox potential was continuously monitored through an ORP electrode (D-75, HORIBA, Japan).

Review form: Reviewer 2

Is the manuscript scientifically sound in its present form?

Yes

Are the interpretations and conclusions justified by the results?

Yes

Is the language acceptable?

Yes

Is it clear how to access all supporting data?

Yes

Do you have any ethical concerns with this paper?

No

Have you any concerns about statistical analyses in this paper?

No

Recommendation?

Accept as is

Comments to the Author(s)

I think the author has revised the manuscript according to my previous comments, and I am satisfied with this revision. I agree to publish this paper in Royal Society Open Science.

Review form: Reviewer 3

Is the manuscript scientifically sound in its present form?

No

Are the interpretations and conclusions justified by the results?

Yes

Is the language acceptable?

Yes

Is it clear how to access all supporting data?

Yes

Do you have any ethical concerns with this paper?

Yes

Have you any concerns about statistical analyses in this paper?

No

Recommendation?

Major revision is needed (please make suggestions in comments)

Comments to the Author(s)

General comments

This study reports an experimental study of removal of selenite and selenate from water by green rust (GR) for understanding of the interaction between selenite/selenate and GR. The authors presented the original research from an experimental study on the synthesized GR and applied this material to the treatment of selenite/selenate in hydrosphere. Whilst the subject of this study is not novel and several earlier studies reported similar investigations, the work of Onoguchi and coworkers investigated the mechanism of sorption and reduction process. I appreciate that the authors have applied EXAFS and various techniques to investigate mechanism. I think that the paper can be improved after revising. However, I recommend a somewhat revision prior to publication in R. Soc. Open Sci.

Specific comments

- 1) The details on TEM-EDS operation parameters should be mentioned, such as accelerating voltage.
- 2) Add specifications of the beamline (energy range of BL5S1, detectors, optical system).
- 3) The detailed analysis procedure of EXAFS should be described in the manuscript.
- 4) P3L52: I think authors made mistake on pKa2 of selenite. The pKa2 of selenite should be 8.32 and this should influence the discussion of selenite sorption and discussion.
- 5) For Fig. 3 and Fig. 4, the GR seems to convert into goethite and magnetite after reaction with Se(IV) under pH 9. However, the dissolved sulfate concentration did not increase. Could authors interpret this? Where is the sulfate?
- 6) The formula of goethite and magnetite should be given in the manuscript.

7) Under pH 8, the sorption mechanisms of selenite/selenate should be different based on Fig. 3. Some discussion can be added in this part.

8) For Fig. 5, why the particle size of reduced Se is different for selenite/selenite?

Decision letter (RSOS-182147.R0)

06-Feb-2019

Dear Professor Tokoro:

Title: Kinetics and mechanism of selenate and selenite removal in solution by green rust-sulfate
Manuscript ID: RSOS-182147

The editor assigned to your paper has now received comments from reviewers. We would like you to revise your paper in accordance with the referee and Subject Editor suggestions which can be found below (not including confidential reports to the Editor). Please note this decision does not guarantee eventual acceptance.

Please submit a copy of your revised paper before 01-Mar-2019. Please note that the revision deadline will expire at 00.00am on this date. If we do not hear from you within this time then it will be assumed that the paper has been withdrawn. In exceptional circumstances, extensions may be possible if agreed with the Editorial Office in advance. We do not allow multiple rounds of revision so we urge you to make every effort to fully address all of the comments at this stage. If deemed necessary by the Editors, your manuscript will be sent back to one or more of the original reviewers for assessment. If the original reviewers are not available we may invite new reviewers.

RSC Associate Editor
Comments to the Author:
(There are no comments.)

Reviewers' Comments to Author:
Reviewer: 2

Comments to the Author(s)
I think the author has revised the manuscript according to my previous comments, and I am satisfied with this revision. I agree to publish this paper in Royal Society Open Science.

Reviewer: 3

Comments to the Author(s)
General comments

This study reports an experimental study of removal of selenite and selenate from water by green rust (GR) for understanding of the interaction between selenite/selenate and GR. The authors presented the original research from an experimental study on the synthesized GR and applied this material to the treatment of selenite/selenate in hydrosphere. Whilst the subject of this study is not novel and several earlier studies reported similar investigations, the work of Onoguchi and coworkers investigated the mechanism of sorption and reduction process. I appreciate that the authors have applied EXAFS and various techniques to investigate mechanism. I think that the paper can be improved after revising. However, I recommend a somewhat revision prior to publication in R. Soc. Open Sci.

Specific comments

- 1) The details on TEM-EDS operation parameters should be mentioned, such as accelerating voltage.
- 2) Add specifications of the beamline (energy range of BL5S1, detectors, optical system).
- 3) The detailed analysis procedure of EXAFS should be described in the manuscript.
- 4) P3L52: I think authors made mistake on pKa2 of selenite. The pKa2 of selenite should be 8.32 and this should influence the discussion of selenite sorption and discussion.

- 5) For Fig. 3 and Fig. 4, the GR seems to convert into goethite and magnetite after reaction with Se(IV) under pH 9. However, the dissolved sulfate concentration did not increase. Could authors interpret this? Where is the sulfate?
- 6) The formula of goethite and magnetite should be given in the manuscript.
- 7) Under pH 8, the sorption mechanisms of selenite/selenate should be different based on Fig. 3. Some discussion can be added in this part.
- 8) For Fig. 5, why the particle size of reduced Se is different for selenite/selenite?

Reviewer: 1

Comments to the Author(s)

This resubmitted article by Onoguchi et al, describes the removal of selenite and selenate through processes involving adsorption or reduction by green rust.

The authors have done a good job improving the language and addressing major reviewer comments. I only have one comment:

Line 55-56: "The total volume was set to reach a total Fe concentration of 0.036 mol/L (2000 mg/L). The Se concentration was set at 500 mg/L while the pH was adjusted to 8 or 9 by adding 0.25 M NaOH. While the synthesis of GR was conducted inside a glovebox, removal experiments were conducted in a separable flask under open air atmosphere, for 2 hours under magnetic stirring (300 rpm) and constant pH. During the experiments, the suspensions were continuously purged with Ar gas to limit the concentration of dissolved oxygen"

I suggest the following alternative text to resolve the confusion with your 'open air' statement:

Removal experiments were conducted by adding the GR suspension to the aqueous solutions containing Se. The suspension was mixed by magnetic stirring (300 rpm) and purged continuously with Ar gas to limit the concentration of dissolved oxygen. The total volume was set to reach a total Fe concentration of 0.036 mol/L (2000 mg/L).

The Se concentration was set at 500 mg/L while the pH was adjusted to 8 or 9 by adding 0.25 M NaOH. An automatic titrator (TS-2000, HIRANUMA SANGYO, Japan) adding 0.5 M NaOH was used to compensate for the pH decrease during the experiments. The redox potential was continuously monitored through an ORP electrode (D-75, HORIBA, Japan).

Author's Response to Decision Letter for (RSOS-182147.R0)

See Appendix D.

Decision letter (RSOS-182147.R1)

12-Mar-2019

Dear Professor Tokoro:

Title: Kinetics and mechanism of selenate and selenite removal in solution by green rust-sulfate
Manuscript ID: RSOS-182147.R1

It is a pleasure to accept your manuscript in its current form for publication in Royal Society Open Science. The chemistry content of Royal Society Open Science is published in collaboration with the Royal Society of Chemistry.

RSC Associate Editor
Comments to the Author:
(There are no comments.)

Reviewer(s)' Comments to Author:

Appendix A**ROYAL SOCIETY
OPEN SCIENCE****Kinetics and mechanism of selenate and selenite removal in
solution by sulfate-green rust**

Journal:	Royal Society Open Science
Manuscript ID	RSOS-181464
Article Type:	Research
Date Submitted by the Author:	30-Sep-2018
Complete List of Authors:	Onoguchi, Aina; Waseda University Faculty of Science and Engineering Granata, Giuseppe; Waseda University Faculty of Science and Engineering Haraguchi, Daisuke; Mitsubishi Material Corporation Central Research Institute Hayashi, Hiroshi; Mitsubishi Material Corporation Central Research Institute Tokoro, Chiharu; Waseda University Faculty of Science and Engineering,
Subject:	Inorganic chemistry < CHEMISTRY
Keywords:	Wastewater treatment, Reduction, Selenium, XAFS
Subject Category:	Chemistry

Kinetics and mechanism of selenate and selenite removal in solution by sulfate-green rust

Aina Onoguchi¹, Giuseppe Granata¹, Daisuke Haraguchi²,
Hiroshi Hayashi², Chiharu Tokoro¹

1 Department of Resources and Environmental Engineering, Waseda University, Okubo 3-4-1, Shinjuku, Tokyo (169-8555)

2 Central Research Institute, Mitsubishi Materials Corporation, 15-2, Fukimatsu, Onahama, Iwaki, Fukushima(971-8101)

Keywords: Wastewater treatment, Reduction, Selenium, XAFS, Removal rate

1. Summary

This work investigated the removal of selenite and selenate from water by sulfate-green rust (GR). Selenite was immobilized by simple adsorption onto GR at pH 8, and by adsorption-reduction at pH 9. Selenate was immobilized by adsorption-reduction to selenite and by direct reduction to Se^0 at both pH 8 and 9. In the process, GR oxidized to goethite and magnetite. The removal of selenite and selenate by GR was kinetically described through a pseudo-second order model based on inductively-coupled plasma spectrometry (ICP) results. XAFS analysis enabled to elucidate the concentration profiles of Se and Fe species in the solid phase and allowed to distinguish two removal mechanism, namely adsorption and reduction. Selenite and selenate were reduced by GR through homogeneous solid-phase reaction upon adsorption and by heterogeneous reaction at the solid-liquid interface. The selenite reduced through heterogeneous reduction with GR was adsorbed onto GR but not reduced further. The redox reaction between GR and selenite/selenate was kinetically described through an irreversible second order bimolecular reaction model based on XAFS concentration profiles. Although the redox reaction became faster at pH 9, simple adsorption was always the fastest removal mechanism.

2. Introduction

Selenium is an essential element for humans but becomes carcinogenic and teratogenic at high concentrations^{1,2}. It commonly exists under four oxidation states, among which Se(IV) as selenite (SeO_3^{2-}) and Se(VI) as selenate (SeO_4^{2-}) are the most stable species in water. Water pollution by selenate and selenite arises mostly from anthropic activities such as agricultural practices³, power generation⁴ and mining⁵. Therefore, given the large amount of water involved in these activities, selenium pollution should not be underestimated. The removal of SeO_4^{2-} and SeO_3^{2-} from water has been object of many research works. Biological methods under anaerobic⁶, oxic⁷ and anoxic⁸ conditions were proven effective and sustainable but relatively slow. Adsorption on carbon nanospheres⁹, ferrous hydroxide¹⁰, ettringite¹¹ and barite¹² were described as possible alternatives to biological methods. However much research is still needed to achieve the efficiency and/or sustainability required to implement real processes. Reduction with zero-valent iron was also proven as an efficient method^{8,13-15}. However, a large amount of zero-valent iron should be specifically added to the wastewater in order to reduce and immobilize selenium species.

*Author for correspondence (tokoro@waseda.jp).

†Present address: Department of Resources and Environmental Engineering, Waseda University, Okubo 3-4-1, Shinjuku, Tokyo (169-8555)

When the contaminated water contains already dissolved iron, an ideal alternative to externally-added removing agents is the immobilization on green rust¹⁶. Green Rust (GR) is a mixed Fe(II)–Fe(III) hydroxide compound with layered double hydroxide (LDH) structure¹⁷. GR can be described through the formula $[\text{Fe}^{\text{II}}_{(1-x)}\text{Fe}^{\text{III}}_x(\text{OH})_2]^{x+}[(x/n)\text{A}_n(\text{m}x/n)\text{H}_2\text{O}]_x$, where A represents anions like chloride, carbonate¹⁸ or sulfate^{19,20}. In GR, Fe(II) and Fe(III) hydroxides units exhibit octahedral structure and occupy positively charged surface portions²¹. Water molecules and anions are instead located in the interlayer between Fe(II)–Fe(III) octahedra to balance the charge²². Due to the presence of Fe(II) on the surface, GR exhibits reducing strength that can be conveniently utilized to reduce oxidized contaminants such as nitrate²³, chromate²⁴ and arsenic anions^{25,26}. GR has also showed an interesting activity in the immobilization of selenate from water. Refait et al. (2000) described the intercalation of selenate into GR's interlayer²⁷. However, authors did not assess the reducing capability of GR and its fate upon intercalation. Hayashi et al. (2009) highlighted that GR can adsorb and reduce selenate to Se^0 , oxidizing itself to goethite and/or magnetite depending on the pH. Authors speculated about the simultaneous reduction of selenate to selenite and Se^0 but did not provide any specific evidence about it²⁸. Moreover, neither the reactions pathway nor the removal kinetics have been elucidated, while there are no information at all about the possibility to remove selenite. This study addressed the removal of selenite and selenate at pH 8 and 9 by sulfate-GR. Sulfate-GR was selected over other types of GRs because the sulfate ion is often available in wastewaters contaminated by selenium (e.g. acid mine drainage). Batch removal experiments were carried out to investigate the evolution of selenium and iron species during removal along with the removal kinetics. An approach based on ICP and XAFS analysis was used to elucidate the evolution of Se(IV), Se(VI) and Fe species, clarify the reactions pathway and assess the contribution of adsorption and reduction to the overall Se removal kinetics.

3. Materials and Methods

3.1 Preparation of Green Rust

GR was prepared within a glove box (UN-650L, UNICO LTD, Japan) under Ar atmosphere to prevent oxidation. All solutions were prepared using distilled and ion exchanged water (Aquarius RFD 240NA, ADVANTEC, JAPAN) and were flushed with Ar gas to remove the dissolved oxygen prior to use. The GR precursor solution was prepared dissolving $\text{FeSO}_4 \cdot 7\text{H}_2\text{O}$ and $\text{Fe}_2(\text{SO}_4)_3 \cdot n\text{H}_2\text{O}$ in the ratio Fe(II)/Fe(III) = 0.75 up to achieve a total Fe concentration of 22.3 g/L. The solution was then titrated with 8 M NaOH (addition speed: $2.3 \times 10^{-7} \text{ dm}^3/\text{min}$) up to reach pH 7.5. GR was separated from the obtained slurry by centrifugation (Himac CR21, HITACHI, JAPAN) at 1000 G for 10 minutes and washed with ion-exchanged-distilled and de-oxygenated water for three times. After washing, the GR precipitate was re-suspended in de-oxygenated water, and later used as it was in removal experiments.

3.2 Selenium removal experiments

Removal experiments were conducted adding the GR suspension to the aqueous solutions containing Se up to reach a total Fe concentration of 2000 mg/L. The Se concentration was set at 500 mg/L while the pH was adjusted to 8 or 9. Experiments were conducted under open air atmosphere, for 2 hours under magnetic stirring (300 rpm) and constant pH. During the experiments, the suspensions were continuously flushed with Ar gas to limit the concentration of dissolved oxygen. An automatic titrator (TS-2000, HIRANUMA SANGYO, Japan) adding 0.5 M NaOH was used to compensate for the pH decrease during the experiments. The redox potential was continuously monitored through an ORP electrode (D-75, HORIBA, Japan).

3.3 Analysis

The concentration of Se in solution was determined by inductively-coupled plasma atomic emission spectrometry (ICP-AES, SPS7800, SEIKO INSTRUMENT, Japan) after filtration. The phase composition and crystal structure of solid products from removal experiments were determined by x-ray diffraction (Geiger flex RAD-IX, RIGAKU, Japan) with a copper target ($\text{CuK}\alpha$) a crystal graphite monochromator and a scintillation detector. The X-ray source was operated at 40 kV and 30 mA with step scanning from 2θ values of 2 to 80° , sequential increments of 0.02° and a scan speed of $2^\circ/\text{min}$. To avoid the oxidation of samples between experiments and analysis, the collected paste samples were mixed with glycerol and transferred to the XRD chamber²⁹. The concentration of Fe(II) was determined through the Phenanthroline method and UV-Vis analysis (DR5400, HACH, USA). The total Fe(T-Fe) was determined by Phenanthroline method upon Fe(III) reduction

with hydroxylammonium chloride. A transmission electron microscope equipped with energy dispersive probe (TEM-EDS, HF-2200, HITACHI, Japan) was utilized to determine size and morphology of solid products. Image mapping was applied to the same micrographs to identify the distribution of Fe and Se in the solid phase. For this purpose, samples were set on an elastic carbon support film (STEM 100 Cu grid, grid pitch 100 μm , OKENSHOJI, Japan) drying directly from the Ar purged glove box after vacuum drying. The Se K-edge and Fe K-edge X-ray adsorption fine structure (XAFS) were performed at the BL5S1 beamline of the Aichi Synchrotron Radiation Center (Aichi Science and Technology Foundation, Japan). The pellets for XAFS analysis were prepared through vacuum drying, grinding, and mixing with boron nitride³⁰ in the Ar purged glove box after solid liquid separation of the slurry.

4. Results and Discussion

4.1 Removal Experiments

The concentration of dissolved selenium in the removal experiments with Se(IV) and Se(VI) at pH 8 and 9 is shown in Fig. 1.

Fig. 1. Se concentration in the removal experiments at pH 8 and 9

Results highlighted that selenite was removed to a larger extent at pH 8, whereas the largest removal of selenate was observed at pH 9. In the experiments with selenite, the Se concentration rapidly decreased upon contact with GR. However, while at pH 8 Se concentration reached a plateau after the first dramatic decrease, at pH 9 selenium re-dissolved. Following this re-dissolution, the concentration of Se in solution started decreasing again at a clearly lower rate. This trend suggested that two different phenomena with probably different rates might be involved in the removal of selenium by GR. In the experiments with Se(VI), Se concentration exhibited similar profiles but with a smoother initial decrease. Furthermore, the best removal performance was observed at pH 9. The larger removal of selenate at pH 9 is contrasting with what reported by Hayashi et al. (2009), who described pH 7.5 as the best condition to remove Se(VI) in the pH range 7-10. To elucidate this aspect, the trends of sulfate concentration and $\text{Fe}^{2+}/\text{total Fe}$ molar ratio ($\text{Fe}^{2+}/\text{T-Fe}$) in solution were continuously monitored during the experiments. Results are shown in Fig.2a and 2b. Fig. 3 shows the redox potential (vs SHE) measured during removal experiments and the cumulative addition of NaOH to titrate the system.

Fig. 2. $[Fe(II)]/[T-Fe]$ molar ratio (a) and SO_4^{2-} release (b) during removal experiments

Fig. 3. Redox potential vs SHE (a) and volume of 0.5 M NaOH added (b) during the removal experiments

The initial $Fe^{2+}/T-Fe$ ratio in GR was 0.66, very close to the ideal ratio reported in literature^{31,32}. This ratio progressively decreased in all experiments, beside the one with Se(IV) at pH 8. **The decreasing $Fe^{2+}/T-Fe$ ratio was a clear suggestion that GR was being oxidized while removing selenite and/or selenate.** Accordingly, SO_4^{2-} was released in solution following GR oxidation (Fig. 2b). Both trends were in general more pronounced at pH 9. In contrast, the $Fe^{2+}/T-Fe$ ratio was nearly constant in the experiment with selenite at pH 8, though the removal of selenium was the largest. This evidence revealed as selenite at pH 8 was removed from solution without reduction. The very low amount of sulfate released in the same experiment can be considered as an evidence that selenite was removed by simple adsorption onto GR. In fact, if the immobilization occurred via intercalation in GR interlayer, the sulfate release should be stoichiometric to selenite in order to maintain the charge balance within GR. Like in the experiments at pH 8, the concentration of dissolved Se decreased rapidly upon contact with GR also at pH 9, thus suggesting a quick adsorption onto GR. However, at pH 9 selenite re-dissolved again after the initial removal. Given the decreasing $Fe^{2+}/T-Fe$ ratio and the release of SO_4^{2-} , it is possible that about 50% of the selenite initially adsorbed was released due to GR oxidation.

The above mentioned consideration were supported also by the redox potential in solution (Fig. 3b). In all experiments, the redox potential sharply decreased upon adding GR to the selenite/selenate solutions. The polarization at lower potentials revealed as the system became electrochemically controlled by the redox couples Fe^{3+}/Fe^{2+} and SeO_4^{2-}/SeO_3^{2-} . Following the initial sharp polarization, the potential started decreasing again but slowly due to reduction of selenite/selenate. The slow decrease was not observed in the experiment with Se(IV) at pH 8, as a further proof that selenite was removed via simple adsorption. In contrast, the largest Eh decrease was observed in the experiments with selenate at pH 9, as also expected from the $Fe^{2+}/T-Fe$ trend. This evidence was a further indication that (i) selenate was easier to reduce than selenite, and that (ii) pH 9 favored the redox reaction.

4.2 GR and Se evolution

The XRD patterns of the solid residues from removal experiments are shown in Fig. 4.

Fig. 4. XRD patterns of the solid residues from removal experiments

The XRD patterns in Fig. 4 exhibited the typical peaks of goethite and magnetite under all investigated conditions except for Se(IV) at pH 8. Although selenium species were not detected, the only presence of GR in the XRD pattern of the solid residue from the experiment with selenite at pH 8 confirmed the removal via adsorption. The presence of Se in all residual solids was confirmed by FE-TEM-EDX analysis and image mapping (Fig. 5). In the TEM micrographs, the hexagonal flakes corresponded to GR²² while the rods observed in the other experiments could be either magnetite or goethite³³, in agreement with XRD results. The solid residue from the experiment with selenite at pH 9 exhibited yellow spherical particles, most likely corresponding to Se⁰. On the other hand, the solid residues from the experiments with selenate exhibited also yellow portions with elongated shape, suggesting the presence of selenite adsorbed onto magnetite and/or goethite.

Fig. 5. TEM micrographs and image mapping of residual solids from experiments with Se(IV) at pH8 (a), Se(IV) at pH9 (b), Se(VI) at pH8 (c), Se(VI) at pH9 (d). (Blue = Fe; Yellow = Se).

4.3 XAFS results

XAFS analysis was conducted in the Se K-edge and Fe K-edge to identify Fe and Se species during removal experiments³⁴. The Se K-edge normalized XANES spectra and the Fe k-edge k^3 weighted EXAFS spectra of solid residues from removal experiments are shown in Fig. 6a and 6b. Results of peak fitting against reference materials are listed in Table 1.

Fig. 6. Normalized Se k-edge XANES spectra (a) and Fe k-edge k^3 weighted EXAFS spectra (b) of end-products against Se(0), Na₂SeO₃, Na₃SeO₄, GR, goethite and magnetite reference materials.

Table 1. Concentration of Se and Fe species in end-products based on XAFS fitting

Condition	Se(VI) [%]	Se(IV) [%]	Se ⁰ [%]	GR [%]	Goethite [%]	Magnetite [%]
Se(IV) pH 8	-	100	-	100	0	0
Se(IV) pH 9	-	8.9	91.1	73.1	26.9	0
Se(VI) pH 8	0	55.4	44.6	70.5	22.3	7.2
Se(VI) pH 9	0	47.1	52.9	17.7	64.0	18.3

The Se k-edge XANES and the EXAFS function $k^3\chi(k)$ spectra of the solid residue from the experiment with Se(IV) at pH 8 revealed the only presence of selenite and GR, as also expected from the above described results. In contrast, Se(IV), Se⁰, GR and goethite were confirmed in the solid residue at pH 9. Fitting results highlighted as about 25% of selenite was reduced to Se⁰ at the expenses of GR, which oxidized to goethite as in (1):

XAFS results from the experiment with selenate at pH 8 exhibited multiple peaks around 12652, 12657 and 12659 eV. These peaks revealed the presence of Se⁰ and Se(IV) in the solid residue. The EXAFS function $k^3\chi(k)$ of the same sample revealed the conversion of about 29% GR to 22% goethite and about 7% magnetite. Clearly, selenate was reduced to selenite and Se⁰ using the electrons generated from the oxidation of GR to goethite and magnetite as in (2)-(5).

Given the solubility of selenite, the presence of Se(IV) in the solid residues revealed as the selenite formed from reduction of selenate was adsorbed by GR and/or iron products. Nevertheless, it was not possible to understand whether selenite was adsorbed upon heterogeneous reduction between GR and selenate, or if it was generated from already adsorbed selenate that remained adsorbed after reduction. This aspect will be discussed in section 3.4.

Increasing the removal pH from 8 to 9 resulted into a more pronounced reduction of selenate to Se⁰. As a consequence of the larger amount of exchanged electrons, more GR oxidized to goethite (64%) and magnetite (18%). The significantly positive effect of pH on the extent of redox reaction is consistent with the stoichiometry of reactions (2)-(5) as hydroxyl ions are required for the reaction to take place. The simultaneous reduction of selenate to both goethite and magnetite is contrasting with results reported by Hayashi et al. (2009) and Myneni

et al., 1997, who described the preferential formation of magnetite at pH 9³⁵. The difference might be due to the different experimental conditions used in this work (e.g. constant pH through addition of NaOH, lower Fe/Se molar ratio).

4.4 Reaction pathway and kinetics

Since the immobilization of selenium involved the adsorption and/or reduction of Se species on GR surface, the kinetic analysis of Se removal was performed using a pseudo-second order model³⁶. According to the model, the adsorption rate is proportional to the second power of the sites available for adsorption/reaction³⁷ as in (6):

$$\frac{dq_t}{dt} = k(q_e - q_t)^2 \quad (6)$$

Where q_e and q_t are the sorption densities at time t and equilibrium, determined respectively as in (7) and (8):

$$q_e = \frac{(C_i - C_e)V}{M} \quad (7)$$

$$q_t = \frac{(C_i - C_t)V}{M} \quad (8)$$

In (7) and (8), C_i , C_e and C_t [mmol/L] are the concentrations of Se(IV) or Se(VI) at initial, equilibrium and time t , respectively. M is the mass of T-Fe [g] calculated from the formula of GR, V is the volume of solution [L] and k is the adsorption rate constant. For the boundary conditions $t = 0$ to $t = t$ and q_t , the integrated form of equation (6) becomes like in (9):

$$\frac{t}{q_t} = \frac{1}{q_e} t + \frac{1}{q_e^2 k} \quad (9)$$

Accordingly, if the removal reaction follows a pseudo-second order kinetics, the experimental points plotted as t/q_t vs time align through a straight line with slope $1/q_e$ and intercept $1/q_e^2 k$. The plots of t/q_t vs time are shown in Fig. 7. Fitting parameters from Fig. 7 are listed in Table 2.

Fig. 7. Fitting of data from removal experiments with pseudo-second order kinetic model

Table 2. Fitting parameters for the lines in Fig. 7 (the rate constant was converted from $g\text{-GR} \times \text{mmol-}Se^{-1} \times \text{min}^{-1}$ to $L \times \text{mmol-}Se^{-1} \times \text{min}^{-1}$)

Condition	q_e [mmol/g]	k [$L \times \text{mmol-}Se^{-1} \times \text{min}^{-1}$]	R^2
Se(IV) pH 8	2.3×10^{-1}	9.9×10^{-1}	1.000
Se(IV) pH 9	1.4×10^{-1}	1.2×10^{-2}	0.980
Se(VI) pH 8	9.6×10^{-2}	4.7×10^{-2}	0.974
Se(VI) pH 9	2.0×10^{-1}	8.6×10^{-2}	0.989

The good fitting from Fig. 7 highlighted the suitability of the pseudo-second order model to describe the removal of selenite and selenate by GR at pH 8 and 9. The removal rate constant was the highest in the experiment with selenite at pH 8, thus highlighting that simple adsorption was the fastest removal mechanism. Increasing the pH from 8 to 9 promoted the redox reaction between selenite and GR. As a consequence, the

kinetics was slower because GR structure was destroyed and the adsorbed selenite was released in solution. In contrast, the removal of selenate was the fastest at pH 9, condition that determined also the most pronounced redox reaction.

To identify the phenomena involved in the removal of selenium and to assess their contribution to the overall kinetics, XAFS analysis was extended to samples at 5, 15, 20, 30, 90 and 120 minutes. Fitting results from XAFS spectra are listed in Table 3 while the plots of data in Table 3 representing the evolution of Se and Fe species through time are shown in Fig. 8. The XAFS spectra are instead provided as supporting information (Fig. S1-S2).

Table 3. Concentration of Se and Fe species in the solid phase (results from XANES and EXAFS fitting)

Condition	Time [min]	Residual Se solution [%]	Se(VI) [%]	Se(IV) [%]	Se ⁰ [%]	GR [%]	Goethite [%]	Magnetite [%]
Se(IV) pH 8	5	66.6	-	33.4	-	100	-	0
	20	60.1	-	39.9	-	100	-	0
	30	58.6	-	41.4	-	100	-	0
	90	58.7	-	41.3	-	100	-	0
	120	58.4	-	41.6	-	100	-	0
Se(IV) pH 9	5	79	-	19	3	97	3	0
	20	89	-	3	9	92	8	0
	30	88	-	2	10	90	10	0
	90	77	-	3	20	80	20	0
	120	72	-	3	26	73	27	0
Se(VI) pH 8	5	94	4	1	1	98	2	0
	20	92	1	4	3	92	6	3
	30	90	0	6	4	87	9	4
	90	84	0	9	7	78	16	6
	120	82	0	10	8	71	22	7
Se(VI) pH 9	5	86	7	3	4	90	5	5
	20	91	2	12	15	67	25	9
	30	90	1	13	16	57	34	9
	90	67	0	16	18	26	58	16
	120	63	0	17	19	18	64	18

Fig. 8. Concentration profiles of Fe and Se species in the solid phase for the experiments with Se(IV) at pH 9 (a-b), Se(VI) at pH 8 (c-d) and Se(VI) at pH 9 (e-f).

From the graphs in Fig. 8, it was clear that selenite was adsorbed onto GR and/or reduced to Se⁰ upon adsorption. The concentration profiles also suggested a fast adsorption of the initial Se(IV) and Se(VI) followed by their (slower) redox reactions to Se⁰ (Fig. 8b) and Se(IV) and/or Se⁰ (fig. 8d and 8f), respectively. The first interesting point to note is the concentration profile of Se(VI) in the solid phase. Although for practical reasons the first sample for XAFS analysis was taken 5 minutes after realizing the contact Se-GR, the Se(VI) concentration must be clearly 0 before contact (time 0). Therefore, the concentration of Se(VI) must have reached a maximum at $t \leq 5$ minutes before being consumed. This was a clear evidence that selenate was quickly adsorbed onto GR, as initially suggested by Fig. 1. Following the adsorption, Se(VI) was reduced to Se(IV) and/or Se⁰. However, Fig. 8b-8d-8f revealed also as the concentrations of reduced selenium species (Se⁰ in the experiment with selenite, Se(IV) and Se⁰ in the experiment with selenate) were still decreasing after all initially adsorbed Se was consumed. This evidence suggested two considerations. First, not only the adsorbed selenite/selenate could be reduced by GR (homogenous reaction) but also their dissolved counterparts upon contact with GR (heterogeneous reaction). Moreover, because selenite is soluble, the presence of Se(IV) in the solid phase could be explained only by considering the adsorption of the selenite formed by heterogeneous reaction between GR and selenate. In this

view, the slow removal of Se following the initial large concentration drop observed in Fig. 1 must be the main consequence of the heterogeneous redox reaction. Another interesting point to note is the simultaneous reduction of selenate to selenite and Se^0 (Fig. 8d-8f). Although the reduction of selenate could be described by a series of two irreversible reductions (e.g. to selenite first and, in turn, to Se^0), the monotonic trend of Se(IV) was more consistent with a parallel reduction of the adsorbed selenate to adsorbed selenite and/or Se^0 . In other words, the adsorbed selenite could not be reduced further to Se^0 . This evidence might be caused by the depletion of Fe(II) in the proximity of the Fe(III) adsorption sites due to reduction of selenate to selenite. A similar consideration can be proposed based on the concentration profiles of Fe species. The monotonic trends of GR, magnetite and goethite suggested that GR oxidized either to magnetite or to goethite, whereas the oxidation of magnetite to goethite did not occur.

Based on GR consumption (XAFS results), amount of precipitated iron, selenium removed (ICP results) and the stoichiometry of redox reactions (2)-(5), an electron balance was carried out to assess the efficiency of GR to reduce Se(IV) and Se(VI) . The relationship between electrons produced through GR oxidation and electrons used in the reduction of Se(IV) and Se(VI) is shown in Fig. 9. The linear correlation with unitary slope shown in Fig. 9 confirmed the effectiveness of GR in the reduction of selenite at pH 9 and selenate at both pH 8 and 9. The slight positive deviation observed from 0.010 mol/dm^3 (condition corresponding to the experimental points at 90 and 120 minutes) in the experiments with Se(VI) at pH 9 revealed that some of the electrons from GR were not used to reduce selenate. There are two possible explanations for the observed deviation. One is the partial unwanted oxidation of GR by air, though Ar gas was continuously bubbled into the reacting system. Another possible explanation involves the reduction of selenate to soluble species such as selenite or selenide³⁸ that were not adsorbed.

Fig. 9. Correlation between electrons for GR oxidation and electrons for Se(IV)/Se(VI) reduction

Given the multiple phenomena occurring simultaneously in the investigated GR-Se systems (e.g. simple adsorption, homogenous/heterogeneous redox reaction), it would be interesting to assess the relationship between them and their contribution on the overall Se removal and kinetics. For this purpose, the kinetics of the redox reaction between GR and selenite/selenate were compared with the above described pseudo-second order model describing the overall removal kinetics. In this work, the kinetics of the redox reactions was determined based on GR concentration profiles from XAFS analysis (Fig. 8). GR concentration profiles were preferred over the concentration profiles of Se(IV) and Se(VI) since the latest ones depend upon adsorption phenomena. The reaction between GR and selenite/selenate was assumed to be a bimolecular first order reaction to both GR and selenite/selenate, thus an overall second order reaction of the type (10):

Unlike previous research works where the concentration of GR was set in large stoichiometric excess to the targeted species³⁹, in this work GR and selenite/selenate were used in the stoichiometric ratio. Under this condition, the second order reaction in (10) can be conveniently treated as a second order reaction to GR⁴⁰. Accordingly:

$$\frac{dC_{\text{GR}}}{dt} = k_{\text{rdx}} C_{\text{GR}}^2 \quad (11)$$

Integration of the second order differential equation in (11) yields to (12):

$$\frac{1}{C_{\text{GR}}} = \frac{1}{C_{\text{GR0}}} + k_{\text{rdx}} t \quad (12)$$

where C_{GR0} and C_{GR} (mmol/L) are the concentrations of GR at time 0 and time t , while k_{rdx} ($L \times mmol^{-1} \times min^{-1}$) is the rate constant of the second order redox reaction between GR and selenite/selenate. Fitting results are shown in Fig. 10. Fitting parameters from Fig. 10 are listed in Table 4.

Fig. 10. Fitting of GR concentration profiles from XAFS results with second order kinetic model

Table 4. Fitting parameters for the lines in Fig. 10

Condition	k_{rdx} [$L \times mmol^{-1} \times min^{-1}$]	R^2
Se(IV) pH 8	-	-
Se(IV) pH 9	5.0×10^{-4}	0.990
Se(VI) pH 8	7.4×10^{-4}	0.999
Se(VI) pH 9	6.3×10^{-3}	0.984

The good correlation of the fitting lines in Fig. 10 confirmed the suitability of the overall second order reaction model to describe the reduction of Se species by GR. Fitting results in Table 4 confirmed as the redox reaction between GR and selenium species became faster at pH 9. The difference between the pseudo-second order removal kinetics in Table 3 and the second-order redox reaction rate constants in Table 4 highlighted that simple adsorption was always the fastest removal mechanism. Comparing the rate parameters also suggested that the immobilization of selenium became slower after GR oxidized. Although the redox reactions are slower than adsorption and do not affect significantly the overall removal kinetics, knowing their rate parameters could be important for the management of end-products. Longer contact times would promote the formation of more crystalline solids (e.g. goethite, Se^0) exhibiting a better solid-liquid separability⁴¹. In contrast, shorter times would lead to a presumably weaker immobilization of selenate by simple adsorption. As a consequence, the selenium immobilized for shorter reaction times could be released more easily from the disposed sludge.

5. Conclusion

This study confirmed the ability of GR to remove selenite and selenate from water, and provided useful information towards the elucidation of the removal mechanism. The immobilization of selenium species by GR occurred through different phenomena depending on pH and Se species. Selenite could be removed from water to a larger extent than selenate. The immobilization occurred *via* simple adsorption at pH 8, and *via* adsorption-reduction with formation of Se^0 and goethite at pH 9. The removal of selenate involved the redox reaction with GR at both pH 8 and 9. In the process, GR oxidized mainly to goethite but also to magnetite, while selenate was reduced to selenite and Se^0 . The concentration profiles from XAFS results suggested that the redox reaction between GR and Se species occurred upon adsorption (homogenous redox reaction) and without adsorption (heterogeneous redox reaction). However, simple adsorption was found to be always the fastest removal mechanism.

Although the redox reaction affects Se removal and removal kinetics to a lesser extent than adsorption, the contact time between GR and selenite/selenate shall be increased to obtain a more efficient immobilization of Se species and the formation more crystalline products.

Data accessibility

All raw data including XRD and XAFS analysis have been uploaded to doi:10.5061/dryad.nk43478. This paper has no additional data.

Competing interests

We declare that we have no competing interests.

Author contribution

A.O. and G.G. gave essential contribution to the experimental acquisition of data and to the drafting of the article. D. H. and H. H. gave substantial contribution to the design of the experiments, analysis and interpretation of data. C. T. gave substantial contribution to conception of the work, interpretation of data and revision of the article as corresponding author. All authors gave final approval for publication.

Funding

This research did not have any official funding support

Ethics

This does not apply to the present study as it does not include humans or animals.

Permission to carry out fieldwork

No fieldwork was carried out in this study

Acknowledgments

A part of the present work was performed within the activities of Research Institute of Sustainable Future Society, Waseda Research Institute for Science and Engineering, Waseda University.

The XAFS analysis were performed using the BL5S1 beamline at the Aichi Synchrotron Radiation Center, Aichi Science & Technology Foundation, Aichi, Japan (Proposal No. 2017P1001).

References

- Ohlendorf HM, Hoffman DJ, Saiki MK, Aldrich TW. 1986. Embryonic mortality and abnormalities of aquatic birds: Apparent impacts of selenium from irrigation drainwater. *Sci Total Environ* 52, 49–63. (10.1016/0048-9697(86)90104-X)
- Clark DR. 1987. Selenium accumulation in mammals exposed to contaminated California irrigation drainwater. *Sci Total Environ* 66, 147–168. (10.1016/0048-9697(87)90084-2)
- Wang D, Alfthan G, Aro A, Lahermo P, Väänänen P. 1994. The impact of selenium fertilisation on the distribution of selenium in rivers in Finland. *Agric Ecosyst Environ* 50, 133–149. (10.1016/0167-8809(94)90132-5)
- Labaran BA, Vohra MS. 2014. Photocatalytic removal of selenite and selenate species: effect of EDTA and other process variables. *Environ Technol* 35, 1091–1100. (10.1080/09593330.2013.861857)
- Zhang P, Sparks DL. 1990. Kinetics of Selenate and Selenite Adsorption/Desorption at the Goethite/Water Interface. *Environ Sci Technol* 24, 1848–1856.
- Oremland RS, Hollibaugh JT, Maest a S, Presser TS, Miller LG, Culbertson CW. 1989. Selenate reduction to elemental selenium by anaerobic bacteria in sediments and culture: biogeochemical significance of a novel, sulfate-independent respiration. *Appl Environ Microbiol* 55, 2333–2343.
- Yoon I-H, Bang S, Kim K-W, Kim MG, Park SY, Choi W-K. 2016. Selenate removal by zero-valent iron in oxic condition: the role of Fe(II) and selenate removal mechanism. *Environ Sci Pollut Res* 23, 1081–1090. (10.1007/s11356-015-4578-4)
- Puranen A, Jonsson M, Dähn R, Cui D. 2009. Immobilization of selenate by iron in aqueous solution under anoxic conditions and the influence of uranyl. *J Nucl Mater* 392, 519–524. (10.1016/J.JNUCMAT.2009.04.016)
- Li M, Wang C, O'Connell MJ, Chan CK. 2015. Carbon nanosphere adsorbents for removal of arsenate and selenate from water. *Environ Sci Nano* 2, 245–250. (10.1039/C4EN00204K)
- Zhang Y, Fu M, Wu D, Zhang Y. 2017. Immobilization of selenite from aqueous solution by structural ferrous hydroxide complexes. *RSC Adv* 7, 13398–13405. (10.1039/C6RA26225B)
- Guo B, Sasaki K, Hirajima T. 2017. Selenite and selenate uptaken in ettringite: Immobilization mechanisms, coordination chemistry, and insights from structure. *Cem Concr Res* 100, 166–175. (10.1016/J.CEMCONRES.2017.07.004)
- Tokunaga K, Takahashi Y. 2017. Effective Removal of Selenite and Selenate Ions from Aqueous Solution by Barite. *Environ Sci Technol* 51, 9194–9021. (10.1021/acs.est.7b01219)
- Liu H, Cai Z, Zhao X, Zhao D, Qian T, Bozack M, et al. 2016. Reductive Removal of Selenate in Water Using Stabilized Zero-Valent Iron Nanoparticles. *Water Environ Res* 88, 694–703. (10.2175/106143016X14609975746929)
- Qin H, Sun Y, Yang H, Fan P, Qiao J, Guan X. 2018. Unexpected effect of buffer solution on removal of selenite and selenate by zerovalent iron. *Chem Eng J* 334, 296–304. (10.1016/J.CEJ.2017.10.025)
- Hu B, Ye F, Jin C, Ma X, Huang C, Sheng G, et al. 2017. The enhancement roles of layered double hydroxide on the reductive immobilization of selenate by nanoscale zero valent iron: Macroscopic and microscopic approaches. *Chemosphere* 184, 408–416. (10.1016/J.CHEMOSPHERE.2017.05.179)
- Refaat P, Simon L, Génin J-MR. 2000. Reduction of SeO4²⁻ Anions and Anoxic Formation of Iron(II)–Iron(III) Hydroxy-Selenate Green Rust. *Environ Sci Technol* 34, 819–825 (10.1021/ES990376G)
- Chaves LHG. 2005. The role of green rust in the environment: a review. *Rev Bras Eng Agrícola e Ambient* 9, 284–8. (10.1590/S1415-43662005000200021)
- Refaat P, Reffass M, Landoulsi J, Sabot R, Jeannin M. 2014. Role of nitrite species during the formation and transformation of the Fe(II-III) hydroxycarbonate green rust. *Colloids Surfaces A Physicochem Eng Asp* 459, 225–232. (10.1016/j.colsurfa.2014.07.004)
- Ahmed IAM, Benning LG, Kakonyi G, Sumoondur AD, Terrill NJ, Shaw S. 2010. Formation of green rust sulfate: A combined in situ time-resolved X-ray scattering and electrochemical study. *Langmuir* 26, 6593–603. (10.1021/la903935j)

20. Mamun A Al, Onoguchi A, Granata G, Tokoro C. 2018. Role of pH in green rust preparation and chromate removal from water. *Appl Clay Sci* 165, 205-213. (<https://doi.org/10.1016/j.clay.2018.08.022>)
21. Mamun A Al, Khin MM, Granata G, Tokoro C. 2018. Removal of chromate from tannery wastewater by sulfate-green rust: case study of the BSCIC tannery estate in Bangladesh. *Resour Process* (In publication).
22. Yin W, Huang L, Pedersen EB, Frandsen C, Hansen HCB. 2017. Glycine buffered synthesis of layered iron(II)-iron(III) hydroxides (green rusts). *J Colloid Interface Sci* 497, 429-438. ([10.1016/j.jcis.2016.11.076](https://doi.org/10.1016/j.jcis.2016.11.076))
23. Choi J, Batchelor B, Won C, Chung J. 2012. Nitrate reduction by green rusts modified with trace metals. *Chemosphere* 86, 860-865. ([10.1016/j.chemosphere.2011.11.035](https://doi.org/10.1016/j.chemosphere.2011.11.035))
24. Legrand L, El Figuigui a, Mercier F, Chausse a. 2004. Reduction of aqueous chromate by Fe(II)/Fe(III) carbonate green rust: Kinetic and mechanistic studies. *Environ Sci Technol* 38, 4587-4595.
25. Wang Y, Morin G, Ona-Nguema G, Juillot F, Guyot F, Calas G, et al. 2010. Evidence for different surface speciation of arsenite and arsenate on green rust: An EXAFS and XANES study. *Environ Sci Technol* 44, 109-115. ([10.1021/es901627e](https://doi.org/10.1021/es901627e))
26. Ona-Nguema G, Morin G, Wang Y, Menguy N, Juillot F, Olivi L, et al. 2009. Arsenite sequestration at the surface of nano-Fe(OH)₂, ferrous-carbonate hydroxide, and green-rust after bioreduction of arsenic-sorbed lepidocrocite by *Shewanella putrefaciens*. *Geochim Cosmochim Acta*, 73, 1359-1381. ([10.1016/j.gca.2008.12.005](https://doi.org/10.1016/j.gca.2008.12.005))
27. Refait P, Simon L, Génin JMR. 2000. Reduction of SeO₄²⁻-anions and anoxic formation of iron(II) - Iron(III) hydroxy-selenate green rust. *Environ Sci Technol* 34, 819-825. ([10.1021/es990376g](https://doi.org/10.1021/es990376g))
28. Hayashi H, Kanie K, Shinoda K, Muramatsu A, Suzuki S, Sasaki H. 2009. pH-dependence of selenate removal from liquid phase by reductive Fe(II)-Fe(III) hydroxysulfate compound, green rust. *Chemosphere* 76, 638-643. ([10.1016/j.chemosphere.2009.04.037](https://doi.org/10.1016/j.chemosphere.2009.04.037))
29. Hansen HCB. 1989. Composition, Stabilization, and Light Absorption of Fe(II)Fe(III) Hydroxy-Carbonate (Green Rust). *Clay Miner* 24, 663-669. ([10.1180/claymin.1989.024.4.08](https://doi.org/10.1180/claymin.1989.024.4.08))
30. Minagawa M, Hisatomi S, Kato T, Granata G, Tokoro C. 2018. Enhancement of copper dissolution by mechanochemical activation of copper ores: Correlation between leaching experiments and DEM simulations. *Adv Powder Technol* 29, 471-478. ([10.1016/J.APT.2017.11.031](https://doi.org/10.1016/J.APT.2017.11.031))
31. Hansen HCB, Guldberg S, Erbs M, Bender Koch C. 2001. Kinetics of nitrate reduction by green rusts-effects of interlayer anion and Fe(II):Fe(III) ratio. *Appl Clay Sci* 18, 81-91. ([10.1016/S0169-1317\(00\)00029-6](https://doi.org/10.1016/S0169-1317(00)00029-6))
32. Ruby C, Abdelmoula M, Naille S, Renard A, Khare V, Ona-Nguema G, et al. 2010. Oxidation modes and thermodynamics of Fe(II)-III oxyhydroxycarbonate green rust: Dissolution-precipitation versus in situ deprotonation. *Geochim Cosmochim Acta* 74, 953-966. ([10.1016/j.gca.2009.10.030](https://doi.org/10.1016/j.gca.2009.10.030))
33. Inoue K, Shinoda K, Suzuki S, Waseda Y. 2010. Oxidation of green rust suspensions containing different chromium ion species. *Corros Sci* 52, 1421-1427. ([10.1016/j.corsci.2009.12.013](https://doi.org/10.1016/j.corsci.2009.12.013))
34. Mamun A Al, Morita M, Matsuoka M, Tokoro C. 2017. Sorption mechanisms of chromate with coprecipitated ferrihydrite in aqueous solution. *J Hazard Mater* 334, 142-9. ([10.1016/J.JHAZMAT.2017.03.058](https://doi.org/10.1016/J.JHAZMAT.2017.03.058))
35. Myneni SC, Tokunaga TK, Brown GE. 1997. Abiotic Selenium Redox Transformations in the Presence of Fe(II,III) Oxides. *Science* 278, 1106-1109. ([10.1126/science.278.5340.1106](https://doi.org/10.1126/science.278.5340.1106))
36. Ho YS, McKay G. 2002. Application of Kinetic Models to the Sorption of Copper(II) on to Peat. *Adsorpt Sci Technol* 20, 797-815. ([10.1260/026361702321104282](https://doi.org/10.1260/026361702321104282))
37. Reddad Z, Gerente C, Andres Y, Cloirec P Le, Cloirec PLE. 2002. Adsorption of Several Metal Ions onto a Low-Cost Biosorbent: Kinetic and Equilibrium Studies Adsorption of Several Metal Ions onto a Low-Cost Biosorbent: Kinetic and Equilibrium Studies. *Environ Sci Technol* 36, 2067-2073. ([10.1021/es0102989](https://doi.org/10.1021/es0102989))
38. Morel F, Hering JG, Morel F. 1993. Principles and applications of aquatic chemistry. Wiley.
39. Aaron GB, Scherer W, Scherer M. 2001. Kinetics of Cr(VI) Reduction by Carbonate Green Rust. *Environ Sci Technol* 35, 3488-3494. ([10.1021/es010579g](https://doi.org/10.1021/es010579g))
40. Levenspiel O. 1976. Chemical reaction engineering. Wiley
41. Davey PT, Scott TR. 1976. Removal of iron from leach liquors by the "goethite" process. *Hydrometallurgy* 2, 25-33.

Appendix B**ROYAL SOCIETY
OPEN SCIENCE****Kinetics and mechanism of selenate and selenite removal in
solution by sulfate-green rust**

Journal:	Royal Society Open Science
Manuscript ID	RSOS-181464
Article Type:	Research
Date Submitted by the Author:	30-Sep-2018
Complete List of Authors:	Onoguchi, Aina; Waseda University Faculty of Science and Engineering Granata, Giuseppe; Waseda University Faculty of Science and Engineering Haraguchi, Daisuke; Mitsubishi Material Corporation Central Research Institute Hayashi, Hiroshi; Mitsubishi Material Corporation Central Research Institute Tokoro, Chiharu; Waseda University Faculty of Science and Engineering,
Subject:	Inorganic chemistry < CHEMISTRY
Keywords:	Wastewater treatment, Reduction, Selenium, XAFS
Subject Category:	Chemistry

Kinetics and mechanism of selenate and selenite removal in solution by sulfate-green rust

Aina Onoguchi¹, Giuseppe Granata¹, Daisuke Haraguchi²,
Hiroshi Hayashi², Chiharu Tokoro¹

1 Department of Resources and Environmental Engineering, Waseda University, Okubo 3-4-1, Shinjuku, Tokyo (169-8555)

2 Central Research Institute, Mitsubishi Materials Corporation, 15-2, Fukimatsu, Onahama, Iwaki, Fukushima(971-8101)

Keywords: Wastewater treatment, Reduction, Selenium, XAFS, Removal rate

1. Summary

This work investigated the removal of selenite and selenate from water by sulfate-green rust (GR). Selenite was immobilized by simple adsorption onto GR at pH 8, and by adsorption-reduction at pH 9. Selenate was immobilized by adsorption-reduction to selenite and by direct reduction to selenite at both pH 8 and 9. In the process, selenite was oxidized to goethite and magnetite. The removal of selenite and selenate by GR was kinetically described through a pseudo-second order model based on inductively-coupled plasma spectrometry (ICP) results. XAFS analysis enabled to elucidate the concentration profiles of Se and Fe species in the solid phase and allowed to distinguish two removal mechanism, namely adsorption and reduction. Selenite and selenate were reduced by GR through homogeneous solid-phase reaction upon adsorption and by heterogeneous reaction at the solid-liquid interface. The selenite reduced through heterogeneous reduction with GR was adsorbed onto GR but not reduced further. The redox reaction between GR and selenite/selenate was kinetically described through an irreversible second order bimolecular reaction model based on XAFS concentration profiles. Although the redox reaction became faster at pH 9, simple adsorption was always the fastest removal mechanism.

2. Introduction

Selenium is an essential element for humans but becomes carcinogenic and teratogenic at high concentrations^{1,2}. It commonly exists under four oxidation states, among which Se(IV) as selenite (SeO_3^{2-}) and Se(VI) as selenate (SeO_4^{2-}) are the most stable species in water. Water pollution by selenate and selenite arises mostly from anthropic activities such as agricultural practices, water generation⁴ and mining⁵. Therefore, given the large amount of water involved in these activities, selenium pollution should not be underestimated. The removal of SeO_4^{2-} and SeO_3^{2-} from water has been the subject of many research works. Biological methods under anaerobic⁶, oxic⁷ and anoxic⁸ conditions were proven effective and sustainable but relatively slow. Adsorption on carbon nanospheres⁹, ferrous hydroxide¹⁰, ettringite¹¹ and barite¹² were described as possible alternatives to biological methods. However much research is still needed to achieve the efficiency and/or sustainability required to implement real processes. Reduction with zero-valent iron was also proven as an efficient method^{8,13-15}. However, a large amount of zero-valent iron should be specifically added to the wastewater in order to reduce and immobilize selenium species.

*Author for correspondence (tokoro@waseda.jp).

†Present address: Department of Resources and Environmental Engineering, Waseda University, Okubo 3-4-1, Shinjuku, Tokyo (169-8555)

When the contaminated water contains already dissolved iron, an ideal alternative to externally-added removing agents is the immobilization on green rust¹⁶. Green Rust (GR) is a mixed Fe(II)–Fe(III) hydroxide compound with layered double hydroxide (LDH) structure¹⁷. GR can be described through the formula $[\text{Fe}^{II}_{(1-x)}\text{Fe}^{III}_x(\text{OH})_2]^{x+}[(x/n)\text{A}_n(\text{m}x/n)\text{H}_2\text{O}]_x$, where A represents anions chloride, carbonate¹⁸ or sulfate^{19,20}. In GR, Fe(II) and Fe(III) hydroxides units exhibit octahedral structure occupy positively charged surface portions²¹. Water molecules and anions are instead located in the interlayer between Fe(II)–Fe(III) octahedra to balance the charge²². Due to the presence of Fe(II) on the surface, GR exhibits reducing strength that can be conveniently utilized to reduce oxidized contaminants such as nitrate²³, chromate²⁴ and arsenic anions^{25,26}. GR has also showed an interesting activity in the immobilization of selenate from water. Refait et al. (2000) described the intercalation of selenate into GR's interlayer²⁷. However, authors did not assess the reducing capability of GR and its fate upon intercalation. Hayashi et al. (2009) highlighted that GR can adsorb and reduce selenate to Se⁰, utilizing itself to goethite and/or magnetite depending on the pH. Authors speculated about the simultaneous reduction of selenate to selenite and Se⁰ but did not provide any specific evidence at it²⁸. Moreover, neither the reactions pathway nor the removal kinetics have been elucidated, while there are no information at all about the possibility to remove selenite. This study addressed the removal of selenite and selenate at 8 and 9 by sulfate-GR. Sulfate-GR was selected for other types of GRs because the sulfate ion is often available in wastewaters contaminated by selenium (e.g. acid mine drainage). Batch removal experiments were carried out to investigate evolution of selenium and iron species during removal along with the removal kinetics. An approach based on ICP and XAFS analysis was used to elucidate the mutation of Se(IV), Se(VI) and Fe species, clarify the reactions pathway and assess the contribution of adsorption and reduction to the overall Se removal kinetics.

3. Materials and Methods

3.1 Preparation of Green Rust

GR was prepared in a glove box (UN-650L, UNICO LTD, Japan) under Ar atmosphere to prevent oxidation. All solutions were prepared using distilled and ion exchanged water (Aquarius RFD 240NA, ADVANTEC, JAPAN) and were flushed with Ar gas to remove the dissolved oxygen prior to use. The GR precursor solution was prepared dissolving $\text{FeSO}_4 \cdot 7\text{H}_2\text{O}$ and $\text{Fe}_2(\text{SO}_4)_3 \cdot n\text{H}_2\text{O}$ in the Fe(II)/Fe(III) = 0.75 up to achieve a total Fe concentration of 2000 mg/L. The solution was then titrated with 8 M NaOH (addition speed: $2.5 \times 10^{-7} \text{ dm}^3/\text{min}$) up to pH 7.5. GR was separated from the obtained slurry by centrifugation (Himac CR21, HITACHI, JAPAN) at 1000 G for 10 minutes and washed with ion-exchanged-distilled and de-oxygenated water for three times. After washing, the GR precipitate was re-suspended in de-oxygenated water, and later used as it was in removal experiments.

3.2 Selenium removal experiments

Removal experiments were conducted adding the GR suspension to the aqueous solutions containing Se up to reach a total Fe concentration of 2000 mg/L. The Se concentration was set at 500 mg/L while the pH was adjusted to 8 or 9. Experiments were conducted under open air atmosphere, for 2 hours under magnetic stirring (300 rpm) and constant pH. During the experiments, the suspensions were continuously flushed with Ar gas to limit the concentration of dissolved oxygen. An automatic titrator (TS-2000, HIRANUMA SANGYO, Japan) adding 0.5 M NaOH was used to compensate for the pH decrease during the experiments. The redox potential was continuously monitored through an ORP electrode (D-75, HORIBA, Japan).

3.3 Analysis

The concentration of Se in solution was determined by inductively-coupled plasma atomic emission spectrometry (ICP-AES, SPS7800, SEIKO INSTRUMENT, Japan) after filtration. The Se composition and crystal structure of solid products from removal experiments were determined by x-ray diffraction (Geiger flex RAD-IX, RIGAKU, Japan) with a copper target ($\text{CuK}\alpha$) a crystal graphite monochromator and a scintillation detector. The X-ray source was operated at 40 kV and 30 mA with step scanning from 2θ values of 2 to 80°, sequential increments of 0.02° and a scan speed of 2°/min. To avoid the oxidation of samples between experiments and analysis, the collected paste samples were mixed with glycerol and transferred to the XRD chamber²⁹. The concentration of Fe(II) was determined through the Phenanthroline method and UV-Vis analysis (DR5400, HACH, USA). The total Fe(T-Fe) was determined by Phenanthroline method upon Fe(III) reduction

with hydroxylammonium chloride. A transmission electron microscope equipped with energy dispersive probe (TEM-EDS, HF-2200, HITACHI, Japan) was utilized to determine size and morphology of solid products. Image mapping was applied to the same micrographs to identify the distribution of Fe and Se in the solid phase. For this purpose, samples were set on an elastic carbon support film (STEM 100 Cu grid, grid pitch 100 μm , OKENSHOJI, Japan) drying directly from the Ar purged glove box after vacuum drying. The Se K-edge and Fe K-edge X-ray adsorption fine structure (XAFS) were performed at the BL551 beamline of the Aichi Synchrotron Radiation Center (Aichi Science and Technology Foundation, Japan). The pellets for XAFS analysis were prepared through vacuum drying, grinding, and mixing with boron nitride³⁰ in the Ar purged glove box after liquid separation of the slurry.

4. Results and Discussion

4.1 Removal Experiments

The concentration of dissolved selenium in the removal experiments with Se(IV) and Se(VI) at pH 8 and 9 is shown in Fig. 1.

Fig. 1. Se concentration in the removal experiments at pH 8 and 9

Results highlighted that selenite was removed to a larger extent at pH 8, whereas the largest removal of selenate was observed at pH 9. In the experiments with selenite, the Se concentration rapidly decreased upon contact with GR. However, while at pH 8 Se concentration reached a plateau after the first dramatic decrease, at pH 9 selenite re-dissolved. Following this re-dissolution, the concentration of Se in solution started decreasing again at a clearly lower rate. This trend suggested that two different phenomena with probably different rates might be involved in the removal of selenium by GR. In the experiments with Se(VI), Se concentration exhibited similar profiles but with a smoother initial decrease. Furthermore, the best removal performance was observed at pH 9. The larger removal of selenate at pH 9 is contrasting with what reported by Hayashi et al. (2009), who described pH 7.5 as the best condition to remove Se(VI) in the pH range 7-10. To elucidate this aspect, the trends of sulfate concentration and $\text{Fe}^{2+}/\text{total Fe}$ molar ratio ($\text{Fe}^{2+}/\text{T-Fe}$) in solution were continuously monitored during the experiments. Results are shown in Fig.2a and 2b. Fig. 3 shows the redox potential (vs SHE) measured during removal experiments and cumulative addition of NaOH to titrate the system.

Fig. 2. $[Fe(II)]/[T-Fe]$ molar ratio (a) and SO_4^{2-} release (b) during removal experiments

Fig. 3. Redox potential vs SHE (a) and volume of 0.5 M NaOH added (b) during the removal experiments

The initial $Fe^{2+}/T-Fe$ ratio in GR was 0.66, very close to the ideal ratio reported in literature^{31,32}. This ratio progressively decreased in all experiments, beside the one with Se(IV) at pH 8. The decreasing $Fe^{2+}/T-Fe$ ratio was a clear suggestion that GR was being oxidized while removing selenite and/or selenate. Accordingly, SO_4^{2-} was released in solution following GR oxidation (Fig. 2b). Both trends were in general more pronounced at pH 9. In contrast, the $Fe^{2+}/T-Fe$ ratio was nearly constant in the experiment with selenite at pH 8, though the removal of selenium was the largest. This evidence revealed as selenite at pH 8 was removed from solution without reduction. The very low amount of sulfate released in the same experiment can be considered as an evidence that selenite was removed by simple adsorption onto GR. In fact, if the immobilization occurred via intercalation in GR interlayer, the sulfate release should be stoichiometric to selenite in order to maintain the charge balance within GR. Like in the experiments at pH 8, the concentration of dissolved Se decreased rapidly upon contact with GR also at pH 9, thus suggesting a quick adsorption onto GR. However, at pH 9 selenite re-dissolved again after the initial removal. Given the decreasing $Fe^{2+}/T-Fe$ ratio and the release of SO_4^{2-} , it is possible that about 50% of the selenite initially adsorbed was released due to GR oxidation.

The above mentioned consideration were supported also by the redox potential in solution (Fig. 3b). In all experiments, the redox potential sharply decreased upon adding GR to the selenite/selenate solutions. The polarization at lower potentials revealed as the system became electrochemically controlled by the redox couples Fe^{3+}/Fe^{2+} and SeO_4^{2-}/SeO_3^{2-} . Following the initial sharp polarization, the potential started decreasing again but slowly due to reduction of selenite/selenate. The slow decrease was not observed in the experiment with Se(IV) at pH 8, as a further proof that selenite was removed via simple adsorption. In contrast, the largest Eh decrease was observed in the experiments with selenate at pH 9, as also expected from the $Fe^{2+}/T-Fe$ trend. This evidence was a further indication that (i) selenate was easier to reduce than selenite, and that (ii) pH 9 favored the redox reaction.

4.2 GR and Se Oxidation

The XRD patterns of the solid residues from removal experiments are shown in Fig. 4.

Fig. 4. XRD patterns of the solid residues from removal experiments

The XRD patterns in Fig. 4 exhibited the typical peaks of goethite and magnetite under all investigated conditions except for Se(IV) at pH 8. Although selenium species were not detected, the only presence of GR in the XRD pattern of the solid residue from the experiment with selenite at pH 8 confirmed the removal via adsorption. The presence of Se in all residual solids was confirmed by FE-TEM-EDX analysis and image mapping (Fig. 5). In the TEM micrographs, the hexagonal flakes corresponded to GR²² while the rods observed in the other experiments could be either magnetite or goethite³³, in agreement with XRD results. The solid residue from the experiment with selenite at pH 9 exhibited yellow spherical particles, most likely corresponding to Se⁰. On the other hand, the solid residues from the experiments with selenate exhibited also low portions with elongated shape, suggesting the presence of selenite adsorbed onto magnetite and/or goethite.

Fig. 5. TEM micrographs and image mapping of residual solids from experiments with Se(IV) at pH8 (a), Se(IV) at pH9 (b), Se(VI) at pH8 (c), Se(VI) at pH9 (d). (Blue = Fe; Yellow = Se).

4.3 XAFS results

XAFS analysis was conducted in the Se K-edge and Fe K-edge to identify Fe and Se species during removal experiments³⁴. The Se K-edge normalized XANES spectra and the Fe k-edge k^3 weighted EXAFS spectra of solid residues from removal experiments are shown in Fig. 6a and 6b. Results of peak fitting against reference materials are listed in Table 1.

Fig. 6. Normalized Se k-edge XANES spectra (a) and Fe k-edge k^3 weighted EXAFS spectra (b) of end-products against Se(0), Na_2SeO_3 , Na_3SeO_4 , GR, goethite and magnetite reference materials.

Table 1. Concentration of Se and Fe species in end-products based on XAFS fitting

Condition	Se(VI) [%]	Se(IV) [%]	Se ⁰ [%]	GR [%]	Goethite [%]	Magnetite [%]
Se(IV) pH 8	-	100	-	100	0	0
Se(IV) pH 9	-	8.9	91.1	73.1	26.9	0
Se(VI) pH 8	0	55.4	44.6	70.5	22.3	7.2
Se(VI) pH 9	0	47.1	52.9	17.7	64.0	18.3

The Se k-edge XANES and the EXAFS function $k^3\chi(k)$ spectra of the solid residue from the experiment with Se(IV) at pH 8 revealed the only presence of selenite and GR, as also expected from the above described results. In contrast, Se(IV), Se⁰, GR and goethite were confirmed in the solid residue at pH 9. Fitting results highlighted as about 25% of selenite was reduced to Se⁰ at the expenses of GR, which oxidized to goethite as in (1):

XAFS results from the experiment with selenate at pH 8 exhibited multiple peaks around 12652, 12657 and 12659 eV. These peaks revealed the presence of Se⁰ and Se(IV) in the solid residue. The EXAFS function $k^3\chi(k)$ of the same sample revealed the conversion of about 29% GR to 22% goethite and about 7% magnetite. Clearly, selenate was reduced to selenite and Se⁰ using the electrons generated from the oxidation of GR to goethite and magnetite as in (2)-(5).

On the solubility of selenite, the presence of Se(IV) in the solid residues revealed as the selenite formed from reduction of selenate was adsorbed by GR and/or iron products. Nevertheless, it was not possible to understand whether selenite was adsorbed upon heterogeneous reduction between GR and selenate, or if it was generated from already adsorbed selenate that remained adsorbed after reduction. This aspect will be discussed in section 3.4.

Increasing the removal pH from 8 to 9 resulted into a more pronounced reduction of selenate to Se⁰. As a consequence of the larger amount of exchanged electrons, more GR oxidized to goethite (64%) and magnetite (18%). The significantly positive effect of pH on the extent of redox reaction is consistent with the stoichiometry of reactions (2)-(5) as hydroxyl ions are required for the reaction to take place. The simultaneous reduction of selenate to both goethite and magnetite is contrasting with results reported by Hayashi et al. (2009) and Myneni

et al., 1997, who described the preferential formation of magnetite at pH 9³⁵. The difference might be due to the different experimental conditions used in this work (e.g. constant pH through addition of NaOH, lower Fe/Se molar ratio).

4.4 Reaction pathway and kinetics

Since the immobilization of selenium involved the adsorption and/or reduction of Se species on GR surface, the kinetic analysis of Se removal was performed using a pseudo-second order model³⁶. According to the model, the adsorption rate is proportional to the second power of the sites available for adsorption/reaction³⁷ as in (6):

$$\frac{dq_t}{dt} = k(q_e - q_t)^2 \quad (6)$$

Where q_e and q_t are the sorption densities at time t and equilibrium, determined respectively as in (7) and (8):

$$q_e = \frac{(C_i - C_e)V}{M} \quad (7)$$

$$q_t = \frac{(C_i - C_t)V}{M} \quad (8)$$

In (7) and (8), C_i , C_e and C_t [mmol/L] are the concentrations of Se(IV) or Se(VI) at initial, equilibrium and time t , respectively. M is the mass of T-Fe [g] calculated from the formula of GR, V is the volume of solution [L] and k is the adsorption rate constant. For the boundary conditions $t = 0$ to $t = t$ and q_t , the integrated form of equation (6) becomes like in (9):

$$\frac{t}{q_t} = \frac{1}{q_e} t + \frac{1}{q_e^2 k} \quad (9)$$

Accordingly, if the removal reaction follows a pseudo-second order kinetics, the experimental points plotted as t/q_t vs time align through a straight line with slope $1/q_e$ and intercept $1/q_e^2 k$. The plots of t/q_t vs time are shown in Fig. 7. Fitting parameters from Fig. 7 are listed in Table 2.

Fig. 7. Fitting of data from removal experiments with pseudo-second order kinetic model

Table 2. Fitting parameters for the lines in Fig. 7 (the rate constant was converted from $g\text{-GR} \times \text{mmol-}Se^{-1} \times \text{min}^{-1}$ to $L \times \text{mmol-}Se^{-1} \times \text{min}^{-1}$)

Condition	q_e [mmol/g]	k [$L \times \text{mmol-}Se^{-1} \times \text{min}^{-1}$]	R^2
Se(IV) pH 8	2.3×10^{-1}	9.9×10^{-1}	1.000
Se(IV) pH 9	1.4×10^{-1}	1.2×10^{-2}	0.980
Se(VI) pH 8	9.6×10^{-2}	4.7×10^{-2}	0.974
Se(VI) pH 9	2.0×10^{-1}	8.6×10^{-2}	0.989

The good fitting from Fig. 7 highlighted the suitability of the pseudo-second order model to describe the removal of selenite and selenate by GR at pH 8 and 9. The removal rate constant was the highest in the experiment with selenite at pH 8, thus highlighting that simple adsorption was the fastest removal mechanism. Increasing the pH from 8 to 9 promoted the redox reaction between selenite and GR. As a consequence, the

kinetics was slower because GR structure was destroyed and the adsorbed selenite was released in solution. In contrast, the removal of selenate was the fastest at pH 9, condition that determined also the most pronounced redox reaction.

To identify the phenomena involved in the removal of selenium and to assess their contribution to the overall kinetics, XAFS analysis was extended to samples at 5, 15, 20, 30, 90 and 120 minutes. Fitting results from XAFS spectra are listed in Table 3 while the plots of data in Table 3 representing the evolution of Se and Fe species through time are shown in Fig. 8. The XAFS spectra are instead provided as supporting information (Fig. S1-S2).

Table 3. Concentration of Se and Fe species in the solid phase (results from XANES and EXAFS fitting)

Condition	Time [min]	Residual Se solution [%]	Se(VI) [%]	Se(IV) [%]	Se ⁰ [%]	GR [%]	Goethite [%]	Magnetite [%]
Se(IV) pH 8	5	66.6	-	33.4	-	100	-	0
	20	60.1	-	39.9	-	100	-	0
	30	58.6	-	41.4	-	100	-	0
	90	58.7	-	41.3	-	100	-	0
	120	58.4	-	41.6	-	100	-	0
Se(IV) pH 9	5	79	-	19	3	97	3	0
	20	89	-	3	9	92	8	0
	30	88	-	2	10	90	10	0
	90	77	-	3	20	80	20	0
	120	72	-	3	26	73	27	0
Se(VI) pH 8	5	94	4	1	1	98	2	0
	20	92	1	4	3	92	6	3
	30	90	0	6	4	87	9	4
	90	84	0	9	7	78	16	6
	120	82	0	10	8	71	22	7
Se(VI) pH 9	5	86	7	3	4	90	5	5
	20	91	2	12	15	67	25	9
	30	90	1	13	16	57	34	9
	90	67	0	16	18	26	58	16
	120	63	0	17	19	18	64	18

Fig. 8. Concentration profiles of Fe and Se species in the solid phase for the experiments with Se(IV) at pH 9 (a-b), Se(VI) at pH 8 (c-d) and Se(VI) at pH 9 (e-f).

From the graphs in Fig. 8, it was clear that selenite was adsorbed onto GR and/or reduced to Se⁰ upon adsorption. The concentration profiles also suggested a fast adsorption of the initial Se(IV) and Se(VI) followed by their (slower) redox reactions to Se⁰ (Fig. 8b) and Se(IV) and/or Se⁰ (fig. 8d and 8f), respectively. The first interesting point to note is the concentration profile of Se(VI) in the solid phase. Although for practical reasons the first sample for XAFS analysis was taken 5 minutes after realizing the contact Se-GR, the Se(VI) concentration must be clearly 0 before contact (time 0). Therefore, the concentration of Se(VI) must have reached a maximum at $t \leq 5$ minutes before being consumed. This was a clear evidence that selenate was quickly adsorbed onto GR, as initially suggested by Fig. 1. Following the adsorption, Se(VI) was reduced to Se(IV) and/or Se⁰. However, Fig. 8b-8d-8f revealed also as the concentrations of reduced selenium species (Se⁰ in the experiment with selenite, Se(IV) and Se⁰ in the experiment with selenate) were still decreasing after all initially adsorbed Se was consumed. This evidence suggested two considerations. First, not only the adsorbed selenite/selenate could be reduced by GR (homogenous reaction) but also their dissolved counterparts upon contact with GR (heterogeneous reaction). Moreover, because selenite is soluble, the presence of Se(IV) in the solid phase could be explained only by considering the adsorption of the selenite formed by heterogeneous reaction between GR and selenate. In this

view, the slow removal of Se following the initial large concentration drop observed in Fig. 1 must be the main consequence of the heterogeneous redox reaction. Another interesting point to note is the simultaneous reduction of selenate to selenite and Se^0 (Fig. 8d-8f). Although the reduction of selenate could be described by a series of two irreversible reductions (e.g. to selenite first and, in turn, to Se^0), the monotonic trend of Se(IV) was more consistent with a parallel reduction of the adsorbed selenate to adsorbed selenite and/or Se^0 . In other words, the adsorbed selenite could not be reduced further to Se^0 . This evidence might be caused by the depletion of Fe(II) in the proximity of the Fe(III) adsorption sites due to reduction of selenate to selenite. A similar consideration can be proposed based on the concentration profiles of Fe species. The monotonic trends of GR, magnetite and goethite suggested that GR oxidized either to magnetite or to goethite, whereas the oxidation of magnetite to goethite did not occur.

Based on GR consumption (XAFS results), amount of precipitated iron, selenium removed (ICP results) and the stoichiometry of redox reactions (2)-(5), an electron balance was carried out to assess the efficiency of GR to reduce Se(IV) and Se(VI) . The relationship between electrons produced through GR oxidation and electrons used in the reduction of Se(IV) and Se(VI) is shown in Fig. 9. The linear correlation with unitary slope shown in Fig. 9 confirmed the effectiveness of GR in the reduction of selenite at pH 9 and selenate at both pH 8 and 9. The slight positive deviation observed from 0.010 mol/dm^3 (condition corresponding to the experimental points at 90 and 120 minutes) in the experiments with Se(VI) at pH 9 revealed that some of the electrons from GR were not used to reduce selenate. There are two possible explanations for the observed deviation. One is the partial unwanted oxidation of GR by air, though Ar gas was continuously bubbled into the reacting system. Another possible explanation involves the reduction of selenate to soluble species such as selenite or selenide³⁸ that were not adsorbed.

Fig. 9. Correlation between electrons for GR oxidation and electrons for Se(IV)/Se(VI) reduction

Given the multiple phenomena occurring simultaneously in the investigated GR-Se systems (e.g. simple adsorption, homogenous/heterogeneous redox reaction), it would be interesting to assess the relationship between them and their contribution on the overall Se removal and kinetics. For this purpose, the kinetics of the redox reaction between GR and selenite/selenate were compared with the above described pseudo-second order model describing the overall removal kinetics. In this work, the kinetics of the redox reactions was determined based on GR concentration profiles from XAFS analysis (Fig. 8). GR concentration profiles were preferred over the concentration profiles of Se(IV) and Se(VI) since the latest ones depend upon adsorption phenomena. The reaction between GR and selenite/selenate was assumed to be a bimolecular first order reaction to both GR and selenite/selenate, thus an overall second order reaction of the type (10):

Unlike previous research works where the concentration of GR was set in large stoichiometric excess to the targeted species³⁹, in this work GR and selenite/selenate were used in the stoichiometric ratio. Under this condition, the second order reaction in (10) can be conveniently treated as a second order reaction to GR⁴⁰. Accordingly:

$$\frac{dC_{\text{GR}}}{dt} = k_{\text{rdx}} C_{\text{GR}}^2 \quad (11)$$

Integration of the second order differential equation in (11) yields to (12):

$$\frac{1}{C_{\text{GR}}} = \frac{1}{C_{\text{GR}0}} + k_{\text{rdx}} t \quad (12)$$

where C_{GR0} and C_{GR} (mmol/L) are the concentrations of GR at time 0 and time t , while k_{rdx} ($L \times mmol^{-1} \times min^{-1}$) is the rate constant of the second order redox reaction between GR and selenite/selenate. Fitting results are shown in Fig. 10. Fitting parameters from Fig. 10 are listed in Table 4.

Fig. 10. Fitting of GR concentration profiles from XAFS results with second order kinetic model

Table 4. Fitting parameters for the lines in Fig. 10

Condition	k_{rdx} [$L \times mmol^{-1} \times min^{-1}$]	R^2
Se(IV) pH 8	-	-
Se(IV) pH 9	5.0×10^{-4}	0.990
Se(VI) pH 8	7.4×10^{-4}	0.999
Se(VI) pH 9	6.3×10^{-3}	0.984

The good correlation of the fitting lines in Fig. 10 confirmed the suitability of the overall second order reaction model to describe the reduction of Se species by GR. Fitting results in Table 4 confirmed as the redox reaction between GR and selenium species became faster at pH 9. The difference between the pseudo-second order removal kinetics in Table 3 and the second-order redox reaction rate constants in Table 4 highlighted that simple adsorption was always the fastest removal mechanism. Comparing the rate parameters also suggested that the immobilization of selenium became slower after GR oxidized. Although the redox reactions are slower than adsorption and do not affect significantly the overall removal kinetics, knowing their rate parameters could be important for the management of end-products. Longer contact times would promote the formation of more crystalline solids (e.g. goethite, Se^0) exhibiting a better solid-liquid separability⁴¹. In contrast, shorter times would lead to a presumably weaker immobilization of selenate by simple adsorption. As a consequence, the selenium immobilized for shorter reaction times could be released more easily from the disposed sludge.

5. Conclusion

This study confirmed the ability of GR to remove selenite and selenate from water, and provided useful information towards the elucidation of the removal mechanism. The immobilization of selenium species by GR occurred through different phenomena depending on pH and Se species. Selenite could be removed from water to a larger extent than selenate. The immobilization occurred *via* simple adsorption at pH 8, and *via* adsorption-reduction with formation of Se^0 and goethite at pH 9. The removal of selenate involved the redox reaction with GR at both pH 8 and 9. In the process, GR oxidized mainly to goethite but also to magnetite, while selenate was reduced to selenite and Se^0 . The concentration profiles from XAFS results suggested that the redox reaction between GR and Se species occurred upon adsorption (homogenous redox reaction) and without adsorption (heterogeneous redox reaction). However, simple adsorption was found to be always the fastest removal mechanism.

Although the redox reaction affects Se removal and removal kinetics to a lesser extent than adsorption, the contact time between GR and selenite/selenate shall be increased to obtain a more efficient immobilization of Se species and the formation more crystalline products.

Data accessibility

All raw data including XRD and XAFS analysis have been uploaded to doi:10.5061/dryad.nk43478. This paper has no additional data.

Competing interests

We declare that we have no competing interests.

Author contribution

A.O. and G.G. gave essential contribution to the experimental acquisition of data and to the drafting of the article. D. H. and H. H. gave substantial contribution to the design of the experiments, analysis and interpretation of data. C. T. gave substantial contribution to conception of the work, interpretation of data and revision of the article as corresponding author. All authors gave final approval for publication.

Funding

This research did not have any official funding support

Ethics

This does not apply to the present study as it does not include humans or animals.

Permission to carry out fieldwork

No fieldwork was carried out in this study

Acknowledgments

A part of the present work was performed within the activities of Research Institute of Sustainable Future Society, Waseda Research Institute for Science and Engineering, Waseda University.

The XAFS analysis were performed using the BL5S1 beamline at the Aichi Synchrotron Radiation Center, Aichi Science & Technology Foundation, Aichi, Japan (Proposal No. 2017P1001).

References

- Ohlendorf HM, Hoffman DJ, Saiki MK, Aldrich TW. 1986. Embryonic mortality and abnormalities of aquatic birds: Apparent impacts of selenium from irrigation drainwater. *Sci Total Environ* 52, 49–63. (10.1016/0048-9697(86)90104-X)
- Clark DR. 1987. Selenium accumulation in mammals exposed to contaminated California irrigation drainwater. *Sci Total Environ* 66, 147–168. (10.1016/0048-9697(87)90084-2)
- Wang D, Alfthan G, Aro A, Lahermo P, Väänänen P. 1994. The impact of selenium fertilisation on the distribution of selenium in rivers in Finland. *Agric Ecosyst Environ* 50, 133–149. (10.1016/0167-8809(94)90132-5)
- Labaran BA, Vohra MS. 2014. Photocatalytic removal of selenite and selenate species: effect of EDTA and other process variables. *Environ Technol* 35, 1091–1100. (10.1080/09593330.2013.861857)
- Zhang P, Sparks DL. 1990. Kinetics of Selenate and Selenite Adsorption/Desorption at the Goethite/Water Interface. *Environ Sci Technol* 24, 1848–1856.
- Oremland RS, Hollibaugh JT, Maest a S, Presser TS, Miller LG, Culbertson CW. 1989. Selenate reduction to elemental selenium by anaerobic bacteria in sediments and culture: biogeochemical significance of a novel, sulfate-independent respiration. *Appl Environ Microbiol* 55, 2333–2343.
- Yoon I-H, Bang S, Kim K-W, Kim MG, Park SY, Choi W-K. 2016. Selenate removal by zero-valent iron in oxic condition: the role of Fe(II) and selenate removal mechanism. *Environ Sci Pollut Res* 23, 1081–1090. (10.1007/s11356-015-4578-4)
- Puranen A, Jonsson M, Dähn R, Cui D. 2009. Immobilization of selenate by iron in aqueous solution under anoxic conditions and the influence of uranyl. *J Nucl Mater* 392, 519–524. (10.1016/J.JNUCMAT.2009.04.016)
- Li M, Wang C, O'Connell MJ, Chan CK. 2015. Carbon nanosphere adsorbents for removal of arsenate and selenate from water. *Environ Sci Nano* 2, 245–250. (10.1039/C4EN00204K)
- Zhang Y, Fu M, Wu D, Zhang Y. 2017. Immobilization of selenite from aqueous solution by structural ferrous hydroxide complexes. *RSC Adv* 7, 13398–13405. (10.1039/C6RA26225B)
- Guo B, Sasaki K, Hirajima T. 2017. Selenite and selenate uptaken in ettringite: Immobilization mechanisms, coordination chemistry, and insights from structure. *Cem Concr Res* 100, 166–175. (10.1016/J.CEMCONRES.2017.07.004)
- Tokunaga K, Takahashi Y. 2017. Effective Removal of Selenite and Selenate Ions from Aqueous Solution by Barite. *Environ Sci Technol* 51, 9194–9021. (10.1021/acs.est.7b01219)
- Liu H, Cai Z, Zhao X, Zhao D, Qian T, Bozack M, et al. 2016. Reductive Removal of Selenate in Water Using Stabilized Zero-Valent Iron Nanoparticles. *Water Environ Res* 88, 694–703. (10.2175/106143016X14609975746929)
- Qin H, Sun Y, Yang H, Fan P, Qiao J, Guan X. 2018. Unexpected effect of buffer solution on removal of selenite and selenate by zerovalent iron. *Chem Eng J* 334, 296–304. (10.1016/J.CEJ.2017.10.025)
- Hu B, Ye F, Jin C, Ma X, Huang C, Sheng G, et al. 2017. The enhancement roles of layered double hydroxide on the reductive immobilization of selenate by nanoscale zero valent iron: Macroscopic and microscopic approaches. *Chemosphere* 184, 408–416. (10.1016/J.CHEMOSPHERE.2017.05.179)
- Refaat P, Simon L, Génin J-MR. 2000. Reduction of SeO₄²⁻ Anions and Anoxic Formation of Iron(II)–Iron(III) Hydroxy-Selenate Green Rust. *Environ Sci Technol* 34, 819–825 (10.1021/ES990376G)
- Chaves LHG. 2005. The role of green rust in the environment: a review. *Rev Bras Eng Agricola e Ambient* 9, 284–8. (10.1590/S1415-43662005000200021)
- Refaat P, Reffass M, Landoulsi J, Sabot R, Jeannin M. 2014. Role of nitrite species during the formation and transformation of the Fe(II-III) hydroxycarbonate green rust. *Colloids Surfaces A Physicochem Eng Asp* 459, 225–232. (10.1016/j.colsurfa.2014.07.004)
- Ahmed IAM, Benning LG, Kakonyi G, Sumoondur AD, Terrill NJ, Shaw S. 2010. Formation of green rust sulfate: A combined in situ time-resolved X-ray scattering and electrochemical study. *Langmuir* 26, 6593–603. (10.1021/la903935j)

20. Mamun A Al, Onoguchi A, Granata G, Tokoro C. 2018. Role of pH in green rust preparation and chromate removal from water. *Appl Clay Sci* 165, 205-213. (<https://doi.org/10.1016/j.clay.2018.08.022>)
21. Mamun A Al, Khin MM, Granata G, Tokoro C. 2018. Removal of chromate from tannery wastewater by sulfate-green rust: case study of the BSCIC tannery estate in Bangladesh. *Resour Process* (In publication).
22. Yin W, Huang L, Pedersen EB, Frandsen C, Hansen HCB. 2017. Glycine buffered synthesis of layered iron(II)-iron(III) hydroxides (green rusts). *J Colloid Interface Sci* 497, 429-438. ([10.1016/j.jcis.2016.11.076](https://doi.org/10.1016/j.jcis.2016.11.076))
23. Choi J, Batchelor B, Won C, Chung J. 2012. Nitrate reduction by green rusts modified with trace metals. *Chemosphere* 86, 860-865. ([10.1016/j.chemosphere.2011.11.035](https://doi.org/10.1016/j.chemosphere.2011.11.035))
24. Legrand L, El Figuigui a, Mercier F, Chausse a. 2004. Reduction of aqueous chromate by Fe(II)/Fe(III) carbonate green rust: Kinetic and mechanistic studies. *Environ Sci Technol* 38, 4587-4595.
25. Wang Y, Morin G, Ona-Nguema G, Juillot F, Guyot F, Calas G, et al. 2010. Evidence for different surface speciation of arsenite and arsenate on green rust: An EXAFS and XANES study. *Environ Sci Technol* 44, 109-115. ([10.1021/es901627e](https://doi.org/10.1021/es901627e))
26. Ona-Nguema G, Morin G, Wang Y, Menguy N, Juillot F, Olivi L, et al. 2009. Arsenite sequestration at the surface of nano-Fe(OH)₂, ferrous-carbonate hydroxide, and green-rust after bioreduction of arsenic-sorbed lepidocrocite by *Shewanella putrefaciens*. *Geochim Cosmochim Acta*, 73, 1359-1381. ([10.1016/j.gca.2008.12.005](https://doi.org/10.1016/j.gca.2008.12.005))
27. Refait P, Simon L, Génin JMR. 2000. Reduction of SeO₄²⁻-anions and anoxic formation of iron(II) - Iron(III) hydroxy-selenate green rust. *Environ Sci Technol* 34, 819-825. ([10.1021/es990376g](https://doi.org/10.1021/es990376g))
28. Hayashi H, Kanie K, Shinoda K, Muramatsu A, Suzuki S, Sasaki H. 2009. pH-dependence of selenate removal from liquid phase by reductive Fe(II)-Fe(III) hydroxysulfate compound, green rust. *Chemosphere* 76, 638-643. ([10.1016/j.chemosphere.2009.04.037](https://doi.org/10.1016/j.chemosphere.2009.04.037))
29. Hansen HCB. 1989. Composition, Stabilization, and Light Absorption of Fe(II)Fe(III) Hydroxy-Carbonate (Green Rust). *Clay Miner* 24, 663-669. ([10.1180/claymin.1989.024.4.08](https://doi.org/10.1180/claymin.1989.024.4.08))
30. Minagawa M, Hisatomi S, Kato T, Granata G, Tokoro C. 2018. Enhancement of copper dissolution by mechanochemical activation of copper ores: Correlation between leaching experiments and DEM simulations. *Adv Powder Technol* 29, 471-478. ([10.1016/J.APT.2017.11.031](https://doi.org/10.1016/J.APT.2017.11.031))
31. Hansen HCB, Guldberg S, Erbs M, Bender Koch C. 2001. Kinetics of nitrate reduction by green rusts-effects of interlayer anion and Fe(II):Fe(III) ratio. *Appl Clay Sci* 18, 81-91. ([10.1016/S0169-1317\(00\)00029-6](https://doi.org/10.1016/S0169-1317(00)00029-6))
32. Ruby C, Abdelmoula M, Naille S, Renard A, Khare V, Ona-Nguema G, et al. 2010. Oxidation modes and thermodynamics of Fe(II)-III oxyhydroxycarbonate green rust: Dissolution-precipitation versus in situ deprotonation. *Geochim Cosmochim Acta* 74, 953-966. ([10.1016/j.gca.2009.10.030](https://doi.org/10.1016/j.gca.2009.10.030))
33. Inoue K, Shinoda K, Suzuki S, Waseda Y. 2010. Oxidation of green rust suspensions containing different chromium ion species. *Corros Sci* 52, 1421-1427. ([10.1016/j.corsci.2009.12.013](https://doi.org/10.1016/j.corsci.2009.12.013))
34. Mamun A Al, Morita M, Matsuoka M, Tokoro C. 2017. Sorption mechanisms of chromate with coprecipitated ferrihydrite in aqueous solution. *J Hazard Mater* 334, 142-9. ([10.1016/J.JHAZMAT.2017.03.058](https://doi.org/10.1016/J.JHAZMAT.2017.03.058))
35. Myneni SC, Tokunaga TK, Brown GE. 1997. Abiotic Selenium Redox Transformations in the Presence of Fe(II,III) Oxides. *Science* 278, 1106-1109. ([10.1126/science.278.5340.1106](https://doi.org/10.1126/science.278.5340.1106))
36. Ho YS, McKay G. 2002. Application of Kinetic Models to the Sorption of Copper(II) on to Peat. *Adsorpt Sci Technol* 20, 797-815. ([10.1260/026361702321104282](https://doi.org/10.1260/026361702321104282))
37. Reddad Z, Gerente C, Andres Y, Cloirec P Le, Cloirec PLE. 2002. Adsorption of Several Metal Ions onto a Low-Cost Biosorbent: Kinetic and Equilibrium Studies Adsorption of Several Metal Ions onto a Low-Cost Biosorbent: Kinetic and Equilibrium Studies. *Environ Sci Technol* 36, 2067-2073. ([10.1021/es0102989](https://doi.org/10.1021/es0102989))
38. Morel F, Hering JG, Morel F. 1993. Principles and applications of aquatic chemistry. Wiley.
39. Aaron GB, Scherer W, Scherer M. 2001. Kinetics of Cr(VI) Reduction by Carbonate Green Rust. *Environ Sci Technol* 35, 3488-3494. ([10.1021/es010579g](https://doi.org/10.1021/es010579g))
40. Levenspiel O. 1976. Chemical reaction engineering. Wiley
41. Davey PT, Scott TR. 1976. Removal of iron from leach liquors by the "goethite" process. *Hydrometallurgy* 2, 25-33.

Appendix C

The manuscript “Kinetics and mechanism of selenate and selenite removal in solution by sulfate-green rust” (RSOS-181464) presents interest study about the Se removal using sulfate-green rust. I would say this study present high quality data to support their conclusion. The manuscript is generally well written and easy to understand. Since the adsorption and reduction of Se by GRs might be complicated, the author tried to make sufficient discussion to reveal the Se removal mechanism. I think this paper could be published with minor revision.

We thank the reviewer for the insightful comments. We addressed them all through modifications of the manuscript and by specific answers here.

Specific Comments:

1. Intorduction P2, L1-2: “When the contaminated water contains already dissolved iron, an ideal alternative to externally-added removing agents is the immobilization on green rust”. I think there should be other reasons to use GRs, such as the easy production using novel methods (*reference 22*) , cheap price but high reducing capacity.

The information have been added as suggested by the reviewer.

2. Preparation of Green Rust: please add some references to support the synthesis methods.

We added a reference to support the synthesis method. We are also open to further specific suggestions if the reviewer has any.

3. Is it possible to merge Fig. 1 and Fig. 2 in one column? They can share the same X axis.

Unfortunately this is very difficult because Fig.2 contains already two figures and it would not be symmetrical if we add a third one.

4. P4, L55: “selenate was easier to reduce that selenite”, “that” should be “than”?

Yes, it was corrected.

5. Fig 8 seems to repeatedly present the data in Table 3.

Table 3 was moved to supporting material.

6. Fig. 10 should be removed since the author provides data in Table 4.

Actually, Table 4 lists the fitting parameters from Fig. 10. Estimating the fitting parameters of Fig. 10 just by looking at it would be impossible.

7. I think the authors have present sufficient data in this study, however, the discussion could be improve by separating the results and discussion, so that the highlights can be emphasized in the discussion. It might be a good way to make the logic clearer when the authors have much data to present.

We agree with the reviewer and added a discussion section accordingly.

Appendix D

Reviewer: 3

Comments to the Author(s)

General comments

This study reports an experimental study of removal of selenite and selenate from water by green rust (GR) for understanding of the interaction between selenite/selenate and GR. The authors presented the original research from an experimental study on the synthesized GR and applied this material to the treatment of selenite/selenate in hydrosphere. Whilst the subject of this study is not novel and several earlier studies reported similar investigations, the work of Onoguchi and coworkers investigated the mechanism of sorption and reduction process. I appreciate that the authors have applied EXAFS and various techniques to investigate mechanism. I think that the paper can be improved after revising. However, I recommend a somewhat revision prior to publication in R. Soc. Open Sci.

We thank the reviewer for taking the time to provide insightful comments that will improve the quality of our manuscript. The answer to his/her comments are provided below while the modifications to the manuscript can be found highlighted in green color.

Specific comments

1) The details on TEM-EDS operation parameters should be mentioned, such as accelerating voltage.

The information about accelerating voltage is now contained in the experimental section.

2) Add specifications of the beamline (energy range of BL5S1, detectors, optical system).

The specifications of the beamline are now included in the experimental section.

3) The detailed analysis procedure of EXAFS should be described in the manuscript.

We added further information about EXAFS analysis.

4) P3L52: I think authors made mistake on pKa2 of selenite. The pKa2 of selenite should be 8.32 and this should influence the discussion of selenite sorption and discussion. Additional considerations about the influence of the pKa were added when answering comment 7.

Yes, we thank the reviewer for noticing it. The pKa2 of selenite was corrected.

5) For Fig. 3 and Fig. 4, the GR seems to convert into goethite and magnetite after reaction with Se(IV) under pH 9. However, the dissolved sulfate concentration did not increase. Could authors interpret this? Where is the sulfate?

Apart from the experiment with Se(IV) at pH 8, where GR was not oxidized and sulfate was not released (Fig. 3b), GR was always partially oxidized. Following the oxidation, a proportional amount of sulfate was released in solution. In fact, the oxidation of GR accounted for about 30% in the experiments with Se(IV) at pH 9 and Se(VI) at pH 9, and for about 80% in the experiment with Se(VI) at pH 9 (Table 1). Accordingly the sulfate release was the largest with Se(VI) at pH 9 (about 200 mg/L), whereas a similar release (about 100 mg/L) was observed in the experiments with Se(IV) at pH 9 and Se(VI) at pH 8. This trend also matched with the observed Fe(II)/FeT (Fig. 3a).

6) The formula of goethite and magnetite should be given in the manuscript.

The formula of goethite and magnetite were added to abstract and manuscript the first time they were mentioned.

7) Under pH 8, the sorption mechanisms of selenite/selenate should be different based on Fig. 3. Some discussion can be added in this part.

The following comment was added to the manuscript to describe the different behavior of selenite and selenate at pH 8.

If the different pH-dependent behaviour can be explained considering that higher pH favour more the redox reactions, the different behaviour of selenite and selenate at the same pH (pH 8) must be explained considering how prone these two species are to adsorption onto GR. Indeed, selenite is smaller than selenate and more negatively charged because completely deprotonated at pH 8. Therefore, a larger removal of selenite by simple adsorption could be somehow expected. In contrast, the reduction of selenate at the same pH was more pronounced, as highlighted by the larger decrease of Fe²⁺/T-Fe ratio and sulfate release. This evidence could be reasonably explained considering the thermodynamic advantage associated with the reduction of a more oxidized anions as SeO₄²⁻.

8) For Fig. 5, why the particle size of reduced Se is different for selenite/selenate?

Since the reduction of selenate was actually faster, a possible explanation for this evidence is the presence of two different reducing mechanisms. It is possible that homogeneous and heterogeneous reductions determined different steric-electrostatic situations around the adsorbed-reduced and reduced particles. As a consequence, the reduced particles grew to a different extent.

We apologize but at this moment we do not have enough information to provide as evidence of this consideration.

Reviewer: 1

Comments to the Author(s)

This resubmitted article by Onoguchi et al, describes the removal of selenite and selenate through processes involving adsorption or reduction by green rust.

The authors have done a good job improving the language and addressing major reviewer comments. I only have one comment:

Line 55-56: "The total volume was set to reach a total Fe concentration of 0.036 mol/L (2000 mg/L). The Se concentration was set at 500 mg/L while the pH was adjusted to 8 or 9 by adding 0.25 M NaOH. While the synthesis of GR was conducted inside a glovebox, removal experiments were conducted in a separable flask under open air atmosphere, for 2 hours under magnetic stirring (300 rpm) and constant pH. During the experiments, the suspensions were continuously purged with Ar gas to limit the concentration of dissolved oxygen"

I suggest the following alternative text to resolve the confusion with your 'open air' statement:

Removal experiments were conducted by adding the GR suspension to the aqueous solutions containing Se. The suspension was mixed by magnetic stirring (300 rpm) and purged continuously with Ar gas to limit the concentration of dissolved oxygen. The total volume was set to reach a total Fe concentration of 0.036 mol/L (2000 mg/L). The Se concentration was set at 500 mg/L while the pH was adjusted to 8 or 9 by adding 0.25 M NaOH. An automatic titrator (TS-2000, HIRANUMA SANGYO, Japan) adding 0.5 M NaOH was used to compensate for the pH decrease during the experiments. The redox potential was continuously monitored through an ORP electrode (D-75, HORIBA, Japan).

We thank the reviewer for suggesting the sentence. The new one is definitely better.